# SCHEMA INFERENCE FOR INTERPRETABLE IMAGE CLASSIFICATION

**Haofei Zhang,**[*] **Mengqi Xue,**[*] **Xiaokang Liu, Kaixuan Chen & Jie Song**[†]
Zhejiang University
{haofeizhang,mqxue,yijiyeah,chenkx,sjie}@zju.edu.cn

**Mingli Song**
Shanghai Institute for Advanced Study, Zhejiang University
brooksong@zju.edu.cn

## ABSTRACT

In this paper, we study a novel inference paradigm, termed as **schema inference**, that learns to deductively infer the explainable predictions by rebuilding the prior deep neural network (DNN) forwarding scheme, guided by the prevalent philosophical cognitive concept of *schema*. We strive to reformulate the conventional model inference pipeline into a graph matching policy that associates the extracted visual concepts of an image with the pre-computed scene impression, by analogy with human reasoning mechanism via impression matching. To this end, we devise an elaborated architecture, termed as *SchemaNet*, as a dedicated instantiation of the proposed *schema inference* concept, that models both the visual semantics of input instances and the learned abstract imaginations of target categories as topological relational graphs. Meanwhile, to capture and leverage the compositional contributions of visual semantics in a global view, we also introduce a universal *Feat2Graph* scheme in *SchemaNet* to establish the relational graphs that contain abundant interaction information. Both the theoretical analysis and the experimental results on several benchmarks demonstrate that the proposed *schema inference* achieves encouraging performance and meanwhile yields a clear picture of the deductive process leading to the predictions. Our code is available at https://github.com/zhfeing/SchemaNet-PyTorch.

## 1 INTRODUCTION

> *"Now this representation of a general procedure of the imagination for providing a concept with its image is what I call the schema for this concept[1]."*
>
> — *Immanuel Kant*

Deep neural networks (DNNs) have demonstrated the increasingly prevailing capabilities in visual representations as compared to conventional hand-crafted features. Take the visual recognition task as an example. The canonical deep learning (DL) scheme for image recognition is to yield an effective visual representation from a stack of non-linear layers along with a fully-connected (FC) classifier at the end (He et al., 2016; Dosovitskiy et al., 2021; Tolstikhin et al., 2021; Yang et al., 2022a), where specifically the inner-product similarities are computed with each category embedding as the prediction. Despite the great success of DL, existing deep networks are typically required to simultaneously perceive low-level patterns as well as high-level semantics to make predictions (Zeiler & Fergus, 2014; Krizhevsky et al., 2017). As such, both the procedure of computing visual representations and the learned category-specific embeddings are opaque to humans, leading to challenges in security-matter scenarios, such as autonomous driving and healthcare applications.

---

[*]Equal contribution.

[†]Corresponding author.

[1]In *Critique of Pure Reason* (A140/B180).

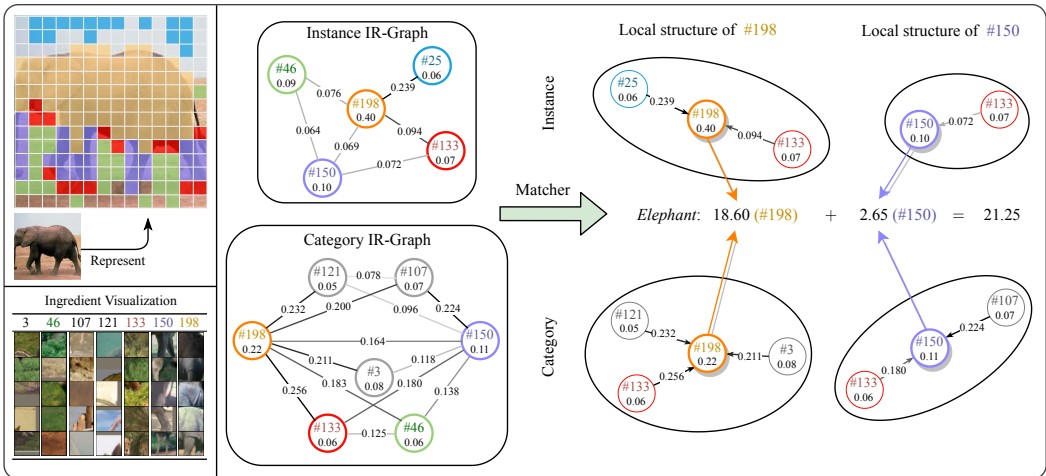

Figure 1: An example showing how an instance IR-Graph is matched to the class imagination. The vertices of IR-Graphs represent visual semantics, and the edges indicate the vertex interactions. The graph matcher captures the similarity between the local structures of joint vertices (*e.g.*, vertex #198 and vertex #150) by aggregating information from their neighbors as class evidences. The final prediction is defined as the sum of all evidence.

Unlike prior works that merely obtain the targets in the black-box manner, here we strive to devise an innovative and generalized DNN inference paradigm by reformulating the traditional one-shot forwarding scheme into an interpretable DNN reasoning framework, resembling what occurs in deductive human reasoning. Towards this end, inspired by the **schema** in Kant's philosophy that describes human cognition as the procedure of associating an image of abstract concepts with the specific sense impression, we propose to formulate DNN inference into an interactive matching procedure between the local visual semantics of an input instance and the abstract category imagination, which is termed as *schema inference* in this paper, leading to the accomplishment of interpretable deductive inference based on visual semantics interactions at the micro-level.

To elaborate the achievement of the proposed concept **schema inference**, we take here image classification, the most basic task in computer vision, as an example to explain our technical details. At a *high level*, the devised *schema inference* scheme leverages a pre-trained DNN to extract *feature ingredients* which are, in fact, the semantics represented by a cluster of deep feature vectors from a specific local region in the image domain. Furthermore, the obtained *feature ingredients* are organized into an *ingredient relation graph* (IR-Graph) for the sake of modeling their interactions that are characterized by the similarity at the semantic-level as well as the adjacency relationship at the spatial-level. We then implement the category-specific imagination as an *ingredient relation atlas* (IR-Atlas) for all target categories induced from observed data samples. As a final step, the *graph similarity* between an instance-level IR-Graph and the category-level IR-Atlas is computed as the measurement for yielding the target predictions. As such, instead of relying on deep features, the desired outputs from schema inference contribute only from the relationship of visual words, as shown in Figure 1.

More specifically, our dedicated *schema*-based architecture, termed as *SchemaNet*, is based on vision Transformers (ViTs) (Dosovitskiy et al., 2021; Touvron et al., 2021), which are nowadays the most prevalent vision backbones. To effectively obtain the feature ingredients, we collect the intermediate features of the backbone from probe data samples clustered by $k$-means algorithm. IR-Graphs are established through a customized *Feat2Graph* module that transfers the discretized ingredients array to graph vertices, and meanwhile builds the connections, which indicates the ingredient interactions relying on the self-attention mechanism (Vaswani et al., 2017) and the spatial adjacency. Eventually, graph similarities are evaluated via a shallow graph convolutional network (GCN).

Our work relates to several existing methods that mine semantic-rich visual words from DNN backbones for self-explanation (Brendel & Bethge, 2019; Chen et al., 2019; Nauta et al., 2021; Xue et al., 2022b; Yang et al., 2022b). Particularly, *BagNet* (Brendel & Bethge, 2019) uses a DNN as visual

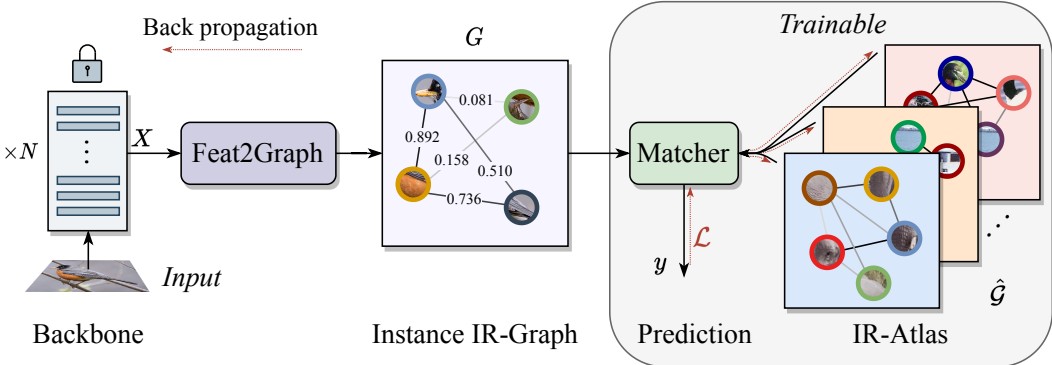

Figure 2: The overall pipeline of our proposed SchemaNet. Firstly, intermediate feature $X$ from the backbone is fed into the *Feat2Graph* module. The converted IR-Graph is then matched to a set of imaginations, *i.e.*, category-level IR-Atlas induced from observed data, for making the prediction.

word extractor that is analogous to traditional bag-of-visual-words (BoVW) representation (Yang et al., 2007) with SIFT features (Lowe, 2004). Moreover, Chen et al. (2019) develop *ProtoPNet* framework that evaluates the similarity scores between image parts and learned class-specific prototypes for inference. Despite their encouraging performance, all these existing methods suffer from the lack of considerations in the compositional contributions of visual semantics, which is, however, already proved critical for both human and DNN inference (Deng et al., 2022), and are not applicable to schema inference in consequence. We discuss more related literature in Appendix A.

In sum, our contribution is an innovative *schema inference* paradigm that reformulates the existing black-box network forwarding into a deductive DNN reasoning procedure, allowing for a clear insight into how the class evidences are gathered and deductively lead to the target predictions. This is achieved by building the dedicated IR-Graph and IR-Atlas with the extracted *feature ingredients* and then performing graph matching between IR-Graph and IR-Atlas to derive the desired predictions. We also provide a theoretical analysis to explain the interpretability of our schema inference results. Experimental results on CIFAR-10/100, Caltech-101, and ImageNet demonstrate that the proposed schema inference yields results superior to the state-of-the-art interpretable approaches. Further, we demonstrate that by transferring to unseen tasks without fine-tuning the matcher, the learned knowledge of each class is, in fact, stored in IR-Atlas rather than the matcher, thereby exhibiting the high interpretability of the proposed method.

## 2 SCHEMANET

In this section, we introduce our proposed SchemaNet in detail. The overall procedure is illustrated in Figure 2, including a Feat2Graph module that converts deep features to instance IR-Graphs, a learnable IR-Atlas, and a graph matcher for making predictions. The main idea is to model an input image as an instance-level graph in which nodes are local semantics captured by the DNN backbone and edges represent their interactions. Meanwhile, a category-level graph is maintained as the imagination for each class driven by training samples. Finally, by measuring the similarity between the instance and category graphs, we are able to interpret how predictions are made.

### 2.1 PRELIMINARY

We first give a brief review of the ViT architecture. The simplest ViT implementation for image classification is proposed by Dosovitskiy et al. (2021), which treats an image as a sequence of independent $16 \times 16$ patches. The embeddings of all patches are then directly fed to a Transformer-based network (Vaswani et al., 2017), *i.e.*, a stack of multiple encoder layers with the same structure: a multi-head self-attention mechanism (MHSA) followed by a multilayer perceptron (MLP) with residual connections. Additionally, they append a class token (CLS) to the input sequence for classification alone rather than the average pooled feature. Following this work, Touvron et al. (2021) propose DeiT appending another distillation token (DIST) to learn soft decision targets from a teacher

model. This paper will mainly focus on DeiT backbones due to the relatively simple yet efficient network architecture.

Let $X \in \mathbb{R}^{(n+\zeta) \times d}$ denotes the input feature (a sequence of $n$ visual tokens and $\zeta$ auxillary tokens, *e.g.*, CLS and DIST, each of which is a $d$-dimensional embedding vector) to a MHSA module. The output can be simplified as $\mathrm{MHSA}(X) = \sum_{h=1}^{H} \tilde{\Psi}_h X W_h$, where $\tilde{\Psi}_h \in \mathbb{R}^{(n+\zeta) \times (n+\zeta)}$ denotes the self-attention matrix normalized by row-wise softmax of head $h \in \{1, \ldots, H\}$, and $W_h \in \mathbb{R}^{d \times d}$ is a learnable projection matrix. As the MLP module processes each token individually, the visual token interactions take place only in the MHSA modules, facilitating the extraction of their relationships from the perspective of ViTs.

For convenience, we define some notations related to the self-attention matrix. Let $\bar{\Psi}$ be the average attention matrix of all heads and $\Psi$ be the symmetric attention matrix: $\Psi = (\bar{\Psi} + \bar{\Psi}^{\top})/2$. We partition $\Psi$ into four submatrices:

$$\Psi = \begin{bmatrix} \Psi^* & \Psi^A \\ \Psi^{A\top} & \Psi^V \end{bmatrix}, \tag{1}$$

where $\Psi^A \in \mathbb{R}^{\zeta \times n}$ is the attention to the auxiliary tokens, $\Psi^V \in \mathbb{R}^{n \times n}$ represents the relationships of visual tokens, and $\Psi^* \in \mathbb{R}^{\zeta \times \zeta}$. Finally, let $\psi^{\mathrm{CLS}} \in \mathbb{R}^n$ represent the attention values to the CLS token extracted from $\Psi^A$.

## 2.2 FEAT2GRAPH

As shown in Figure 2, given a pre-trained backbone and an input image, the intermediate feature $X$ is converted to an IR-Graph $G$ for inference by the Feat2Graph module which includes three steps: (1) discretizing $X$ into a sequence of feature ingredients with specific semantics; (2) mapping the ingredients to weighted vertices of IR-Graph; (3) assigning a weighted edge to each vertex pair indicating their interaction. We start by defining IR-Graph and IR-Atlas mathematically.

**Definition 1** (IR-Graph). *An IR-Graph is an undirected fully-connected graph $G = (E, V)$, in which vertiex set $V$ is the set of feature ingredients for an instance or a specific category, and edge set $E$ encodes the interactions of vertex pairs.*

In particular, to indicate the vertex importance (for instance, "bird head" should contribute to the prediction of bird more than "sky" in human cognition), a non-negative weight $\lambda \in \mathbb{R}_+$ is assigned to each vertex. Let $\Lambda \in \mathbb{R}_+^{|V|} = \{\lambda_i\}_{i=1}^{|V|}$ be the collection of all vertex weights. Besides, the interaction between vertex $i$ and $j$ is quantified by a non-negative weight $e_{i,j} \in \mathbb{R}_+$. With a slight abuse of notation, we define $E \in \mathbb{R}_+^{|V| \times |V|}$ as the weighted adjacency matrix in the following sections.

The IR-Atlas is defined to represent the imagination of all categories:

**Definition 2** (IR-Atlas). *An IR-Atlas $\hat{\mathcal{G}}$ of $C$ classes is a set of category-level IR-Graphs, in which element $\hat{G}_c = (\hat{E}_c, \hat{V}_c)$ for class $c$ has learnable vertex weights $\hat{\Lambda}_c$ and edge weights $\hat{E}_c$.*

**Discretization.**    The main purpose of feature discretization is to mine the most common patterns appearing in the dataset, each of which corresponds to a human understandable semantic, named visual word, such as "car wheel" or "bird head". However, local regions with the same semantic may show differently because of scaling, rotation, or even distortion. In our approach, we utilize the DNN backbone to extract a relatively uniform feature instead of the traditional SIFT feature utilized in (Yang et al., 2007). To be specific, a visual vocabulary $\Omega = \{\omega_i\}_{i=1}^M$ ($\omega_i \in \mathbb{R}^d$) of size $M$ is constructed by $k$-means clustering running on the collection $\mathbb{X}$ of visual tokens extracted from the probe dataset[2]. Further, a deep feature $X$ (CLS and DIST are removed) is discretized to a sequence of feature ingredients by replacing each element $x$ with the index of the closest visual word

$$\mathrm{Ingredient}(x) = \underset{i \in \{1, \ldots, M\}}{\arg \min} \|x - \omega_i\|_2. \tag{2}$$

Strictly, the *ingredient* is referred to as the *index* of visual word $\omega$. With the entire ingredient set $\mathbb{M} = \{1, \ldots, M\}$, the discretized sequence is denoted as $\tilde{X} = (\tilde{x}_1, \ldots, \tilde{x}_n)$, where $\tilde{x}_i \in \mathbb{M}$ is computed from Equation (2). The detailed settings of $M$ are presented in Appendix E.1.

---

[2]For probe dataset with $D$ instances, the collection $\mathbb{X}$ has $n \times D$ visual tokens (ignoring CLS and DIST).

**Feat2Vertex.** After we have computed the discretized feature $\tilde{X}$, the vertices of this feature are the unique ingredients: $V = \text{Unique}(\tilde{X})$. The importance of each vertex is measured from two criterions: the contribution to the DNN's final prediction and the appearance statistically. Formally, for vertex $v \in \mathbb{M}$, the importance is defined as

$$\lambda_v = \alpha_1 \lambda_v^{\text{CLS}} + \alpha_2 \lambda_v^{\text{bag}} = \alpha_1 \sum_{i \in \Xi(v|\tilde{X})} \psi_i^{\text{CLS}} + \alpha_2 |\Xi(v|\tilde{X})|, \tag{3}$$

where $\Xi(v|\tilde{X})$ is the set of all the appeared position of ingredient $v$ in $\tilde{X}$, $\psi_i^{\text{CLS}}$ is the attention between the $i$-th visual token and the CLS token, and $\alpha_{1,2} \geq 0$ are learnable weights balancing the two terms. When $\alpha_1 = 0$, $\Lambda$ is equivalent to BoVW representation.

**Feat2Edge.** With the self-attention matrix $\Psi^V$ extracted from Equation (1), the interactions between vertices can be defined and computed with efficiency. For any two different vertices $u, v \in V$, the edge weight is the comprehensive consideration of the similarity in the view of ViT and spatial adjacency. The first term is the average attention between all repeated pairs $\Pi[(u,v)|\tilde{X}]$:

$$e_{u,v}^{\text{attn}} = \frac{1}{\Pi[(u,v)|\tilde{X}]} \sum_{(i,j) \in \Pi[(u,v)|\tilde{X}]} \Psi_{i,j}^V, \tag{4}$$

where $\Pi[(u,v)|\tilde{X}]$ is the Cartesian product of $\Xi(u|\tilde{X})$ and $\Xi(v|\tilde{X})$, and $\Psi_{i,j}^C$ is the attention between the visual tokens at $i$ and $j$ positions. Furthermore, we define the adjacency as

$$e_{u,v}^{\text{adj}} = \frac{1}{\Pi[(u,v)|\tilde{X}]} \sum_{(i,j) \in \Pi[(u,v)|\tilde{X}]} \frac{1}{\epsilon + \|\text{Pos}(i) - \text{Pos}(j)\|_2}, \tag{5}$$

where function $\text{Pos}(\cdot)$ returns the original 2D coordinates of the input visual token with regard to the patch array after the patch embedding module of ViTs. Eventually, the interaction between vertices $u$ and $v$ is the weighted sum of the two components with learnable $\beta_{1,2}$:

$$e_{u,v} = \beta_1 e_{u,v}^{\text{attn}} + \beta_2 e_{u,v}^{\text{adj}}. \tag{6}$$

Equations (4) and (5) are both invariant when exchanging vertex $u$ and $v$, so our IR-Graph is equivalent to undirected graph.

It is worth noting that only *semantics* and *their relationships* are preserved in IR-Graphs for further inference rather than deep features.

## 2.3 MATCHER

After converting an input image to IR-Graph, the matcher finds the most similar category-level graph in IR-Atlas. The overall procedure is shown in Figure 3, which is composed of a GCN module and a similarity computing module (Sim) that generates the final prediction. Detailed settings of the matcher are in Appendix E.3.

For feeding IR-Graphs to the GCN module, we assign each ingredient $m \in \mathbb{M}$ (*i.e.*, graph vertices) with a trainable embedding vector, each of which is initialized from an $d_G$-dimensional random vector drawn independently from multivariate Gaussian distribution $\mathcal{N}(\mathbf{0}, I_{d_G})$.

In the GCN module, we adopt GraphConv (Morris et al., 2019) with slight modifications for weighted edges. Let $F \in \mathbb{R}^{|V| \times d_G}$ be the input feature of all vertices to a GraphConv layer, the output is computed as

$$\text{GraphConv}(F) = \text{Norm}\left(\sigma\left((I_{d_G} + E)FW\right)\right), \quad (7)$$

where $\sigma$ denotes a non-linear activation function, Norm denotes feature normalization, and $W \in \mathbb{R}^{d_G \times d_G}$ is a

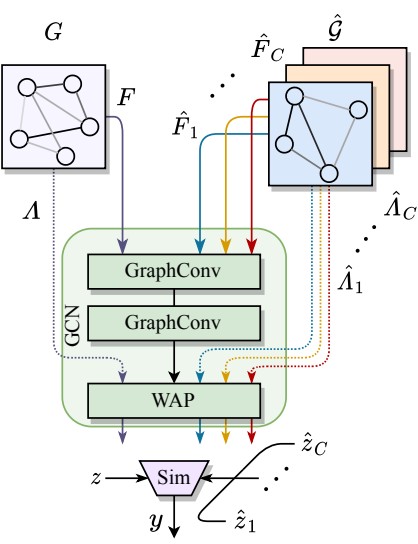

Figure 3: Illustration of the matcher.

learnable projection matrix. After passing through total $L_G$ layers of GraphConv, a weighted average pooling (WAP) layer summarizes the vertex embeddings $F^{(L_G)}$ weighted by $\Lambda$, yielding the graph representation $z = \Lambda F^{(L_G)}$. Now we have $z$ for $G$ and $\hat{Z} \in \mathbb{R}^{C \times d_G} = (\hat{z}_1, \ldots, \hat{z}_C)$ for all graphs in $\hat{\mathcal{G}}$. By computing the inner-product similarities, the final prediction logits is defined as $y = \hat{Z}z^\top$.

**Prediction interpretation.** Given an instance graph $G$ and a category graph $\hat{G}$, we now analyze how the prediction is made based on their vertices and edges. We start with analyzing the similarity score $s = \hat{z}z^\top$, where $z = \Lambda F^{(L_G)}$ and $\hat{z} = \hat{\Lambda}\hat{F}^{(L_G)}$ are weighted sum of the vertex embeddings respectively. For the sake of further discussion, let $f_v^{(l)}$ denotes the embedding vector of vertex $v$ output from the $l$-th GraphConv layer, and $\lambda_v^{(l)}$ denotes its weight. The original vertex embedding of $v$ is defined as $f_v^{(0)}$, which is identical for the same ingredient in any two graphs. Besides, let $\Phi = V \cap \hat{V}$ be the set of shared vertices in $G$ and $\hat{G}$. The computation of $s$ can be expanded as

$$s = \sum_{(u,v) \in \hat{V} \times V} \hat{\lambda}_u \lambda_v \hat{f}_u^{(L_G)} f_v^{(L_G)\top}. \tag{8}$$

Since directly analyzing Equation (8) is hard, we show an approximation and proof it in Appendix B.

**Theorem 1.** *For a shallow GCN module, Equation (8) can be approximated by*

$$s = \sum_{\phi \in \Phi} \hat{\lambda}_\phi \lambda_\phi \hat{f}_\phi^{(L_G)} f_\phi^{(L_G)\top}. \tag{9}$$

*Particularly, if $L_G = 0$ and $\alpha_1 = 0$, our method is equivalent to BoVW with a linear classifier.*

Further, as delineated in Corollary 1, the interpretability of the graph matcher with a shallow GCN can be stated as: (1) the final prediction score for a category is the summation of all present class evidence (shared vertices $\phi$ in $\Phi$) represented by the vertex weights; (2) the shared neighbors (local structure) connected to the shared vertex $\phi$ also contribute to the final prediction.

## 2.4 TRAINING SCHEMANET

Except for training SchemaNet by cross-entropy loss $\mathcal{L}_{CE}$ between the predictions and ground truths, we further constrain the complexity of IR-Atlas. More formally, the complexity is defined for both edges and vertex weights of all graphs in IR-Atlas as

$$\mathcal{L}_v = \frac{1}{C} \sum_{c=1}^{C} \mathcal{H}(\hat{\Lambda}_c), \quad \mathcal{L}_e = \frac{1}{C|\hat{V}|} \sum_{c=1}^{C} \sum_{u \in \hat{V}} \mathcal{H}(\hat{E}_{c,u}), \tag{10}$$

where function $\mathcal{H}(x)$ computes the entropy of the input vector $x \in \mathbb{R}_+^k$ normalized by its sum of $k$ components, and $\hat{E}_{c,u}$ denotes the weighted edges connected to vertex $u$ in $\hat{G}_c$ of class $c$. The final optimization goal is

$$\mathcal{L} = \mathcal{L}_{CE} + \gamma_v \mathcal{L}_v + \gamma_e \mathcal{L}_e, \tag{11}$$

where $\gamma_v$ and $\gamma_e$ are hyperparameters. The overall training procedure is shown in Appendix C.

**Sparsification.** Each graph in IR-Atlas is initialized as a fully-connected graph with random vertex and edge weights. However, this requires $\mathcal{O}(C|V|^2)$ memory space for storage and training the whole set of edges. To alleviate this issue, we initialize IR-Atlas by averaging the instance IR-Graphs for each class and remove the edges connected to vertices whose weights are below a given threshold $\delta_t = 0.01$ from the category-specific graph. Such a procedure will not only dramatically decrease the learnable parameters but also boost the final performance, as shown in Table 1.

## 3 EXPERIMENTS

### 3.1 IMPLEMENTATION

**Datasets.** We evaluate our method on CIFAR-10/100 (Krizhevsky et al., 2009), Caltech-101 (Li et al., 2022), and ImageNet (Deng et al., 2009). Particularly, Caltech-101 has around 9k images, and we manually split it into a training set with 7.4k images and a test set with 1.3k images.

Table 1: Comparison results on CIFAR-10/100 and Caltech-101 with different backbones. We report the top-1 accuracy, number of learnable parameters, and FLOPs. The results in gray color are merely for listing the accuracy of ViTs rather than for making comparisons.

| Backbone | Method | CIFAR-10 | | | CIFAR-100 | | | Caltech-101 | | |
|---|---|---|---|---|---|---|---|---|---|---|
| | | Acc | #param. | FLOPs | Acc | #param. | FLOPs | Acc | #param. | FLOPs |
| - | BoVW-SIFT | 20.20 | - | - | - | - | - | - | - | - |
| DeiT-Tiny | Base | 96.69 | 5.53M | 1.27G | 82.88 | 5.54M | 1.27G | 92.53 | 5.54M | 1.27G |
| | Backbone-FC | 94.58 | 1.93K | 1.06G | 76.74 | 19.3K | 1.06G | 76.71 | 19.5K | 1.06G |
| | BoVW-Deep | 94.95 | 10.3K | 1.06G | 75.45 | 103K | 1.06G | 60.82 | 104K | 1.06G |
| | BagNet | 70.27 | 5.73M | 1.10G | 42.90 | 5.84M | 1.10G | 72.54 | 5.85M | 1.10G |
| | SchemaNet | 95.92 | 396K | 1.19G | 77.11 | 105M | 2.40G | 83.24 | 106M | 2.40G |
| | SchemaNet-Init | 95.96 | 234K | 1.06G | 78.45 | 573K | 1.06G | 87.57 | 564K | 1.06G |
| DeiT-Small | Base | 97.77 | 21.7M | 4.63G | 87.65 | 21.7M | 4.63G | 94.96 | 21.7M | 4.63G |
| | Backbone-FC | 96.62 | 3.85K | 3.87G | 82.44 | 38.5K | 3.87G | 86.94 | 38.9K | 3.87G |
| | BoVW-Deep | 95.61 | 10.3K | 3.89G | 77.39 | 103K | 3.89G | 65.54 | 104K | 3.89G |
| | BagNet | 83.90 | 22.1M | 3.95G | 50.49 | 22.2M | 3.95G | 78.13 | 22.2M | 3.95G |
| | SchemaNet | 97.42 | 396K | 4.00G | 82.21 | 105M | 5.21G | 84.66 | 106M | 5.21G |
| | SchemaNet-Init | 97.35 | 235K | 3.89G | 82.46 | 591K | 3.89G | 90.09 | 589K | 3.89G |
| DeiT-Base | Base | 98.41 | 85.8M | 17.6G | 89.17 | 85.9M | 17.6G | 95.83 | 85.9M | 17.6G |
| | Backbone-FC | 97.04 | 7.69K | 14.7G | 81.66 | 76.9K | 14.7G | 88.20 | 77.7K | 14.7G |
| | BoVW-Deep | 95.50 | 10.3K | 14.7G | 72.23 | 103K | 14.7G | 66.17 | 104K | 14.7G |
| | BagNet | 90.71 | 86.6M | 14.9G | 67.84 | 86.8M | 14.9G | 86.55 | 86.8M | 14.9G |
| | SchemaNet | 97.26 | 396K | 14.8G | 79.26 | 105M | 16.1G | 81.12 | 106M | 16.1G |
| | SchemaNet-Init | 97.07 | 235K | 14.7G | 79.36 | 606K | 14.7G | 90.72 | 593K | 14.7G |

**Selection of the hyperparameters.** Several hyperparameters are involved in our method, including $\lambda_{v,e}$ in Equation (11) for adjusting the sparsity of IR-Atlas. We set $\lambda_v = 0.5$ and $\lambda_e = 0.75$ as the default value for the following evaluation and the sensitive analyses are presented in Appendix H. The initial values of learnable $\alpha_{1,2}$ and $\beta_{1,2}$ are set to 0.5, and we show their learning curves in Figure 9.

## 3.2 EXPERIMENTAL RESULTS

We evaluate our method and comparison baselines with the following settings:

- **BoVW-SIFT:** the traditional BoVW approach that utilizing SIFT feature for constructing the visual vocabulary (Yang et al., 2007).

- **Base:** the base ViT model directly trained on the benchmark datasets with initial weights obtained from the official repository in (Touvron et al., 2021), which is our backbone.

- **Backbone-FC:** the frozen backbone with an FC layer. The intermediate features extracted from the frozen backbone are then fed to a global average pooling layer followed by a linear classifier, similar to the standard CNN protocol.

- **BoVW-Deep:** the BoVW approach with our extracted visual vocabulary.

- **BagNet:** the implementation of BagNet (Brendel & Bethge, 2019) with ViT backbone which is constructed following our "Base" setting.

- **SchemaNet:** our proposed SchemaNet without initialization.

- **SchemaNet-Init:** our proposed SchemaNet initialized by the average instance IR-Graphs.

The comparison results are shown in Table 1, where we demonstrate the top-1 accuracy, the number of learnable parameters, as well as the FLOPs for each setting. It is noticeable that the proposed SchemaNet consistently outperforms the baseline methods. Specifically, for the backbone of DeiT-Tiny, ours achieves significant improvement (*i.e.*, about 25.7% and 35.3% absolute gain on CIFAR-10 and CIFAR-100, respectively) over BagNet. With a larger backbone such as DeiT-Small and DeiT-Base, though the performance gap slightly shrinks, our SchemaNet with initializations still yields results superior to BagNet, by an average absolute gain of 12.7%. The reason is that, unlike BagNet that uses summation to obtain the similarity map as the BoVW representation (without feature discretization), "BoVW-Deep" is implemented with the discretized visual words. As such, both of

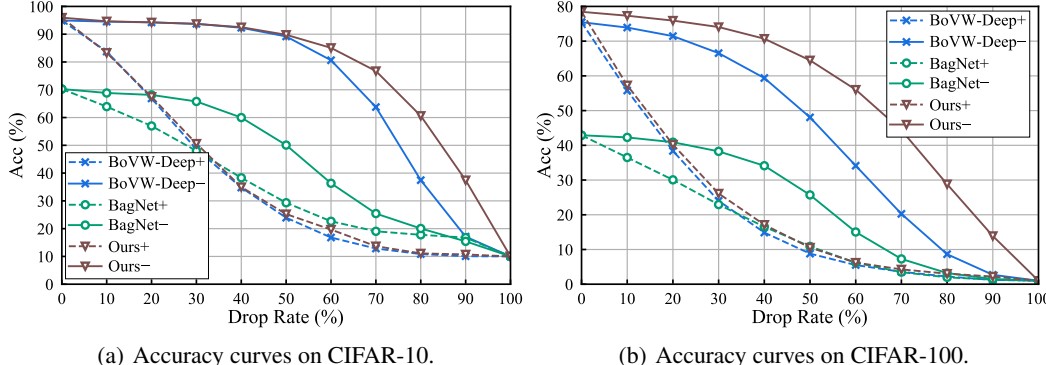

(a) Accuracy curves on CIFAR-10.

(b) Accuracy curves on CIFAR-100.

Figure 4: Accuracy decay curves of perturbation tests. The curve name ending with "+"/"−" represents positive/negative perturbation, accordingly.

the approaches do not consider the visual word interactions, leading to the inferior performance especially when the size of the visual vocabulary expands on CIFAR-100 and Caltech-101.

Moreover, compared with "Backbone-FC" that merely relies on intermediate deep features, the proposed SchemaNet also demonstrates encouraging results, achieving about a 1.4% absolute gain. Furthermore, when initialization and redundant vertices are removed as indicated in Section 3.1, our "SchemaNet-Init", with an even less learnable number of parameters, still leads to a higher accuracy. The proposed method also delivers gratifying performance, on par with the prevalent deep ViT especially on CIFAR-10, but gives a clear insight on the reasoning procedure.

We present more experimental results in Appendix F, including the comparison results on ImageNet (Appendix F.1), employing other attribution methods (Appendix F.2), attacking with adversarial images (Appendix F.3), the extendability analysis of our SchemaNet (Appendix F.4), and the inference cost of all components (Appendix F.5).

## 3.3 EVALUATION OF THE INTERPRETABILITY

To figure out how interpretability contributes to the model prediction, we adopt the positive and negative perturbation tests presented in (Chefer et al., 2021) to give both quantitative and qualitative evaluations on the interpretability. For a fair comparison, the attention values to the CLS token $\phi^{\text{CLS}}$ are extracted as the pixel relevance for BoVW-Deep, BagNet, and SchemaNet-Init with the DeiT-Tiny backbone. During the testing stage, we gradually drop the pixels in relevance descending/ascending order (for positive and negative perturbation, respectively) and measure the top-1 accuracy of the models. We plot the accuracy curves in Figure 4, showing that: (1) in the positive perturbation test, our method behaves similarly to BoVW while outperforming BagNet; (2) in the negative perturbation test, our method achieves better performance by a large margin (the area-under-the-curve of ours is about 5.71% higher than BagNet on CIFAR-10, and 11.23% higher than BagNet on CIFAR-100).

## 3.4 VISUALIZATION

In Figure 5, we show the examples including instance IR-Graphs and learned IR-Atlas with DeiT-Tiny backbone trained on Caltech-101 dataset (for visualization purposes, the number of ingredients is set to 256, which are shown entirely in Figure 11). Each row shows several instances of a specific class. The first column includes the category graphs and excerpts of appeared ingredients in Figure 11 for quicker reference. We further render the vertices, *i.e.*, ingredients, in the category graph with different colors, and the appeared ingredients in the instance images and graphs are colored uniformly corresponding to the category graph.

We interpret the visualization from three perspectives: the interpretability of the ingredients, the interpretability of the edges, and the consistency between instance and category graphs. (1) Thanks to the powerful representation learning capability of DNNs, the extracted ingredients are able to represent explicit semantics (such as "fuselage", "bird legs", etc.) with robustness. Besides, the

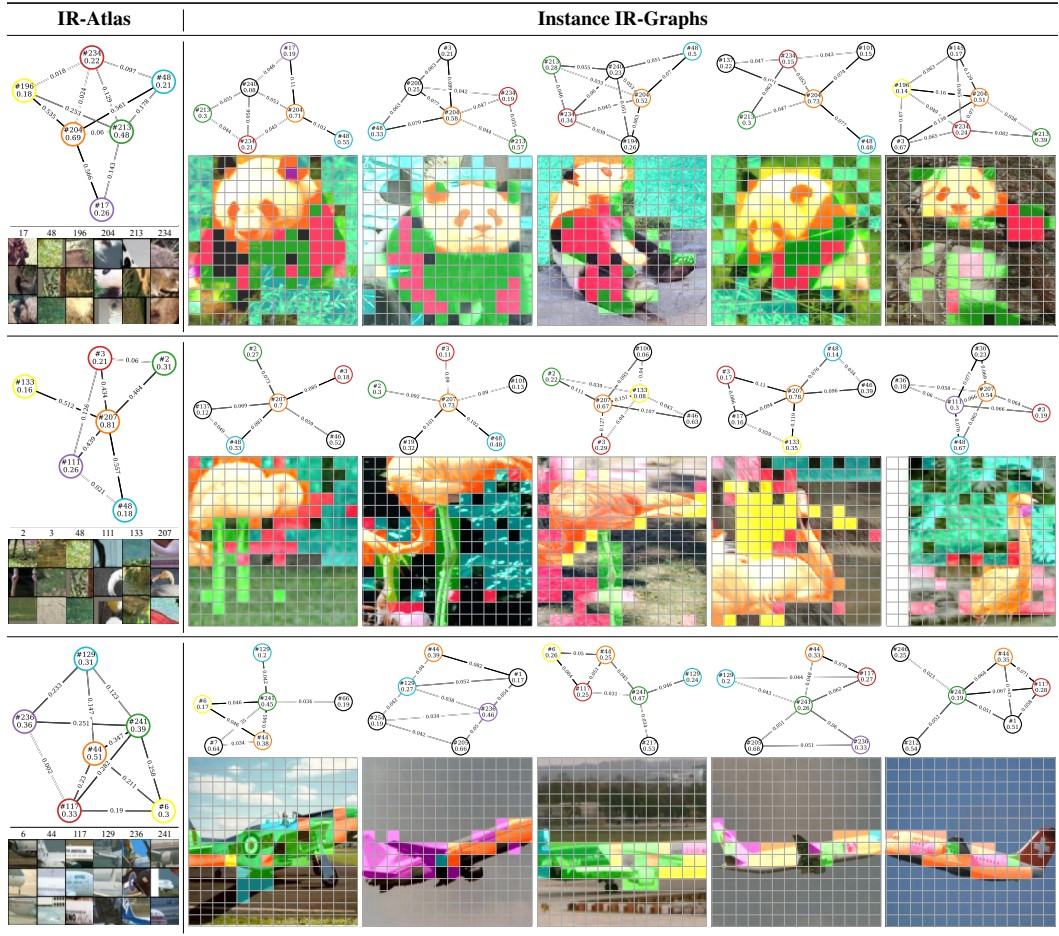

Figure 5: Examples of the learned IR-Atlas and instance IR-Graphs randomly sampled from three categories ("panda", "flamingo", and "airplanes") on Caltech-101. Please zoom in for a better view.

learned weight of each vertex is also consistent with human intuition as the object ingredients get higher weights than the background ones. (2) The learned edge weights tend to connect object parts to the adjacent background and other object parts while ignoring the background connections, which helps distinguish fine-grain categories with the perception of the surroundings. (3) As the category graphs are driven by the instances, they learn to capture and model the general character of instances and eventually form the abstract imaginations of all categories. More examples are in Appendix I.

## 4 CONCLUSION AND OUTLOOK

In this paper, we propose a novel inference paradigm, named *schema inference*, guided by Kant's philosophy towards resembling human deductive reasoning of associating the abstract concept image with the specific sense impression. To this end, we reformulate the traditional DNN inference into a graph matching scheme by evaluating the similarity between instance-level IR-Graph and category-level imagination in a deductive manner. Specifically, the graph vertices are visual semantics represented by common feature vectors from DNN's intermediate layer. Besides, the edges indicate the vertex interactions characterized by semantic similarity and spatial adjacency, which facilitate capturing the compositional contributions to the predictions. Theoretical analysis and experimental results on several benchmarks demonstrate the superiority and interpretability of schema inference. In future work, we will implement schema inference to more complicated vision tasks, such as visual question answering, that enables linking the visual semantics to the phrases in human language, achieving a more powerful yet interpretable reasoning paradigm.

## ACKNOWLEDGMENTS

This work is supported by National Natural Science Foundation of China (61976186, U20B2066, 62106220), Ningbo Natural Science Foundation (2021J189), the Starry Night Science Fund of Zhejiang University Shanghai Institute for Advanced Study (Grant No. SN-ZJU-SIAS-001), Open Research Projects of Zhejiang Lab (NO. 2019KD0AD01/018) and the Fundamental Research Funds for the Central Universities (2021FZZX001-23).

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

# A  ADDITIONAL RELATED WORK

In this section, we introduce literature that is related to our method.

## A.1  VISUAL CONCEPT ANALYSIS

Visual concept analysis aims to interpret the behavior of pre-trained deep models by extracting and tracking intuitive visual concepts during the DNN's inference procedure. Previous works use masks (Zhang et al., 2020b), explanatory graphs (Zhang et al., 2020a) and probabilistic models (Konforti et al., 2020) to interpret the internal layers of convolutional neural networks (CNNs) (Liu et al., 2022; Yu et al., 2023). Further, Deng et al. (2022) propose to capture and analyze the interaction of visual concepts contributing cooperatively to the prediction results for CNNs. However, they use Shapley values to compute the interaction, which is computationally expensive. Consequently, we adopt vision Transformers (ViTs) (Dosovitskiy et al., 2021; Touvron et al., 2021) as the DNN backbone of our method instead of conventional CNNs thanks to the multi-head self-attention (MHSA) mechanism (Vaswani et al., 2017) that explicitly encodes the interactions of visual tokens in the similarity level.

More recently, VRX (Ge et al., 2021) is proposed to use a graphical model to interpret the prediction of a pre-trained DNN, in which edges are constructed to represent the spatial relation of visual semantics. They further utilize a GCN module to make a prediction with the input structural concept graphs by training to mimic DNN's prediction. Nevertheless, their approach requires training a graph for every input sample, which is extremely expensive for applying to large-scale datasets. Besides, VRX does not construct the class imagination that carries the category-specific knowledge, meaning that the GCN predictor must learn to memorize and distinguish all the class-specific graphs, which impairs the overall interpretability. Our proposed schema inference, however, explicitly creates an interactive matching procedure between the instances and category imagination. Moreover, our GNN model only captures the local structure for gathering class evidence, which is easier to accomplish.

## A.2  VISUAL CONCEPT LEARNING

Visual concept learning refers to a range of methods that mine semantic-rich visual words from a DNN, which are further utilized to make interpretable predictions towards a self-explanatory model in reality. Concept bottleneck models (CBMs) (Koh et al., 2020; Zarlenga et al., 2022; Deng et al., 2022; Wong & McPherson, 2021) link the neurons with human-interpretable semantics explicitly, encouraging the trustworthiness of DNNs. Human interventions of learned bottleneck layers can fix the misclassified concepts to improve the model performance. Another line of visual concept learning, part-prototype-based methods, collectively makes predictions on target tasks with DNN and semantic-rich prototypes, which can be divided into two schools according to their prototypes. (1) Bag-of-Visual-Word (BoVW) approaches (Brendel & Bethge, 2019; Gidaris et al., 2020; Tripathi et al., 2022) obtain prototypes with semantics through passively clustering the hand-crafted features (Yang et al., 2007) or deep features into a set of discrete embeddings (visual words) that relate to specific visual semantics. The prediction procedure of these approaches is analog to the BoW model in natural language processing (NLP), in that an interpretable representation is constructed based on the statistics of the occurrence of the visual words, and then fed to an interpretable classifier, such as the linear classifier or decision trees. (2) In contrast, part-prototypical networks (ProtoPNets) (Chen et al., 2019) and the following works (Nauta et al., 2021; Xue et al., 2022a; Zhang et al., 2022; Peters, 2022; Rymarczyk et al., 2021) jointly train DNN (as the feature extractor) and parameterized prototypes. Then decisions are made based on the linear combination of similarity scores between prototypes and feature vectors at all the spatial locations. Despite the relatively high performance, as discussed in (Brendel & Bethge, 2019; Hoffmann et al., 2021), the similarity of the learned prototypes and deep features in the embedding space may be significantly different in the input space. As such, we only compare those approaches in which prototypes are generated based on clustering for the same level of interpretability. To the best of our knowledge, none of the existing works have explored a deductive inference paradigm based on the interaction of visual semantics.

### A.3 SCENE GRAPH GENERATION

A scene graph can be generated from an input image to excavate the collection of objects, attributes, and relationships in a whole scene. Typical graph-based representations and learning algorithms (Herzig et al., 2018; Chang et al., 2021; Shit et al., 2022; Zhong et al., 2021; Zareian et al., 2020) adopt graph neural networks (GNNs) (Kipf & Welling, 2016; Jing et al., 2021a; Yang et al., 2020b; Jing et al., 2021b; Yang et al., 2020a; Jing et al., 2022; Yang et al., 2019) to model the relationships between objects in a scene. Similar to scene graphs, our proposed SchemaNet also utilizes GNNs to represent the relationships between ingredients related to local semantics. However, the nodes in GNNs of SchemaNet represent more fine-grained representations, *i.e.*, local semantics of objects, compared to those relating to the whole objects in scene graphs. Moreover, in SchemaNet, the trained GNN is responsible for estimating the similarity between an instance IR-Graph and category IR-Graphs and making classification based on the similarity scores.

### A.4 FEATURE ATTRIBUTION FOR EXPLAINABLE AI

In the "Feat2Vertex" module, we extract the attention to the CLS token as one important component of the ingredient importance. Currently, plenty of approaches have been proposed for indicating local relevance (forming saliency maps) to the DNN's prediction, termed feature attribution methods. Most existing works can be roughly divided into three classes: perturbation-based, gradient-based, and decomposition-based approaches. Perturbation-based approaches (Strumbelj & Kononenko, 2010; Zeiler & Fergus, 2014; Ancona et al., 2019; Zintgraf et al., 2017) compute attribution by evaluating output differences via removing or altering input features. Gradient-based approaches (Simonyan et al., 2014; Shrikumar et al., 2017; Sundararajan et al., 2017; Selvaraju et al., 2017; Feng et al., 2022) compute gradients *w.r.t.* the input feature through backpropagation. Decomposition-based approaches (Bach et al., 2015; Montavon et al., 2017; Chefer et al., 2021) propagate the final prediction to the input following the Deep Taylor Decomposition (Montavon et al., 2017). Besides, CAM (Zhou et al., 2016) and ABN (Fukui et al., 2019) provide interpretable predictions with a learnable attribution module. As most previous methods focus on CNNs, Chefer et al. (2021) propose ViT-LRP tailored for vision Transformers. However, most of the methods mentioned above are designed to generate a saliency map for a particular class, making them inefficient in our implementation due to the $\mathcal{O}(C)$ computational complexity or dependency on gradients.

## B PROOF OF THEOREM 1

We first restate the theorem:

**Theorem** (Theorem 1 restated). *For a shallow GCN module, Equation* (8) *can be approximated by*

$$s = \sum_{\phi \in \Phi} \hat{\lambda}_\phi \lambda_\phi \hat{f}_\phi^{(L_G)} f_\phi^{(L_G)\top}. \tag{12}$$

*Particularly, if $L_G = 0$ and $\alpha_1 = 0$, our method is equivalent to BoVW with a linear classifier.*

For further discussion, we first prove the following lemma:

**Lemma 1.** *For random vectors $f$, $g \in \mathbb{R}^{d_G}$ drawn **independently** from multivariate Gaussian distribution $\mathcal{N}(\mathbf{0}, I)$, we have*

$$\begin{aligned}
\mathbb{E}_{f,g}\left[fW^\top W g^\top\right] &= 0 \\
\mathbb{E}_f\left[fW^\top W f^\top\right] &= \|W\|_F^2,
\end{aligned} \tag{13}$$

*where $\|\cdot\|_F$ is the matrix Frobenius norm, and $W \in \mathbb{R}^{d_G \times d_G}$ is a projection matrix.*

*Proof.* To begin with, we expand term $fW^\top W g^\top$ as

$$fW^\top W g^\top = \sum_{r=1}^{d_G} \sum_{s=1}^{d_G} \sum_{t=1}^{d_G} f_s g_t w_{r,s} w_{r,t}. \tag{14}$$

Therefore, the expectation

$$\mathbb{E}_{f,g}\left[fW^\top W g^\top\right] = \sum_{r=1}^{d_G} \sum_{s=1}^{d_G} \sum_{t=1}^{d_G} w_{r,s} w_{r,t} \mathbb{E}_{f_s}\left[f_s\right] \mathbb{E}_{g_t}\left[g_t\right] = 0, \tag{15}$$

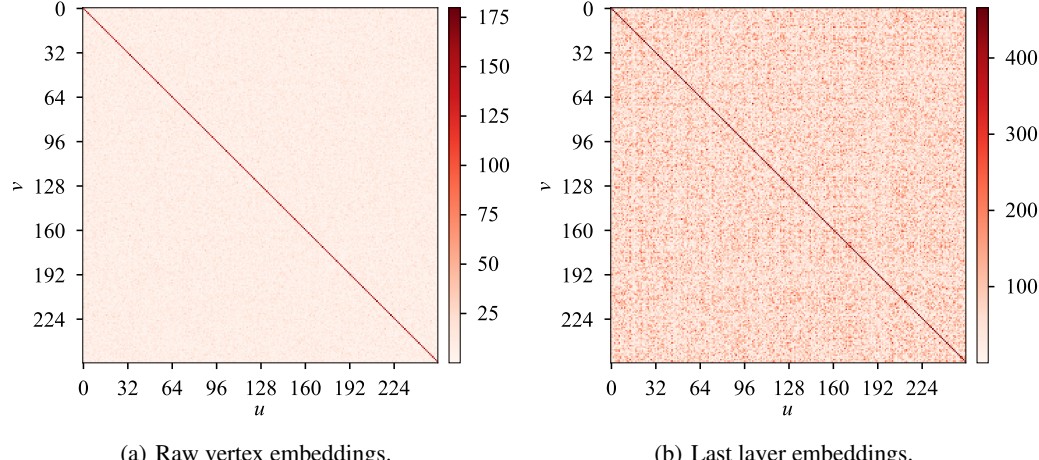

(a) Raw vertex embeddings. (b) Last layer embeddings.

Figure 6: Vertex embedding similarities between a vertex $v$ in the instance IR-Graph and a vertex $u$ in the category IR-Atlas, which are extracted from the original vertex embeddings and output from the last GraphConv layer. We show the average similarity over 1000 samples on the Caltech-101 dataset with a visual vocabulary size of 256.

while

$$
\begin{aligned}
\mathbb{E}_f\left[fW^\top W f^\top\right] &= \sum_{r=1}^{d_G} \mathbb{E}_f\left[\sum_{s=1}^{d_G}\sum_{t=1}^{d_G} w_{r,s}w_{r,t}f_s f_t\right] \\
&= \sum_{r=1}^{d_G} \mathbb{E}_f\left[\sum_{s=1}^{d_G} w_{r,s}^2 f_s^2\right] \\
&= \sum_{r=1}^{d_G}\sum_{s=1}^{d_G} w_{r,s}^2 = \|W\|_F^2 .
\end{aligned}
\tag{16}
$$

□

Particularly, if $W$ is an identity matrix

$$
\mathbb{E}_f\left[fW^\top W f^\top\right] = d_G .
\tag{17}
$$

Further, if the components of matrix $W$ are drawn i.i.d. from Gaussian distribution $\mathcal{N}(0,1)$ and are independent with $f$,

$$
\mathbb{E}_{f,W}\left[fW^\top W f^\top\right] = d_G^2 .
\tag{18}
$$

Now we proof Theorem 1.

*Proof.* We start with a simple case that the depth of GCN module is zero.

As $G$ and $\hat{G}$ share the same original vertex embeddings, we have

$$
s = \sum_{(u,v)\in \hat{V}\times V} \hat{\lambda}_u \lambda_v \hat{f}_u^{(0)} f_v^{(0)\top} = \sum_{(u,v)\in \hat{V}\times V} \hat{\lambda}_u \lambda_v f_u^{(0)} f_v^{(0)\top} .
\tag{19}
$$

According to Lemma 1, the expectation of the same vertex term $\mathbb{E}\left[ff^\top\right] = d_G$ is more significant than different vertices terms $\hat{\lambda}_u \lambda_v f_u^{(0)} f_v^{(0)\top}$ for $u \neq v$ when $G$ and $\hat{G}$ are constrained with sparsity. In Figure 6(a), we visualize the vertex similarities $f_u^{(0)} f_v^{(0)\top}$, showing that the similarities between different vertices are significantly lower than that of the same vertex.

Consequently, Equation (8) in the case of $L_G = 0$ can be simplified as

$$s = \sum_{\phi \in \Phi} \hat{\lambda}_\phi \lambda_\phi \|f_\phi^{(0)}\|_2^2, \tag{20}$$

where $\Phi = V \cap \hat{V}$ is the set of shared vertices of $G$ and $\hat{G}$.

With an extra assumption that $\alpha_1 = 0$, $\lambda_\phi$ therefore represents the count of visual word $\phi$. Combing the learnable term $\hat{\lambda}_\phi \|f_\phi^{(0)}\|_2^2$ as $w_\phi$, the prediction of the matcher is

$$y = \Lambda W_{\text{FC}}, \tag{21}$$

where $W_{\text{FC}} \in \mathbb{R}^{|V| \times C}$ is the learnable projection matrix of a linear classifier.

Now, we prove the whole theorem. For different vertices $u \in \hat{V}$ and $v \in V$, the summation term $s_{u,v} = \hat{\lambda}_u \lambda_v \hat{f}_u^{(L_G)} f_v^{(L_G)^\top}$ can be rewritten as the aggregation form with feature output from the $(L_G - 1)$-th layer:

$$s_{u,v} = \hat{\lambda}_u \lambda_v \sum_{(i,j) \in \dot{\mathcal{N}}_{\hat{G}}(u) \times \dot{\mathcal{N}}_G(v)} \hat{e}_{u,i} e_{v,j} \hat{f}_i^{(L_G-1)} W^{(L_G)} W^{(L_G)^\top} f_j^{(L_G-1)^\top}, \tag{22}$$

where $\dot{\mathcal{N}}_{\hat{G}}(u)$ represent the neighbors of vertex $u$ in Graph $\hat{G}$ and with $u$ itself, and $e_{u,u} = 1$ as the self loop.

With the assumption that $\hat{f}_i^{(L_G-1)}$ and $f_j^{(L_G-1)}$ are whitened random vectors (we say a random vector $x \in \mathbb{R}^d$ is whitened if all the components are zero mean and independent from each other, which can be achieved by normalization), the summation term $\hat{f}_i^{(L_G-1)} W^{(L_G)} W^{(L_G)^\top} f_j^{(L_G-1)^\top}$ with vertex $i = j$ will be significantly larger than those with different vertices. We further illustrate the vertex similarities output from the final layer in Figure 6(b), showing that $\hat{f}_u^{(L_G)} f_v^{(L_G)^\top}$ is conspicuous for vertices $u = v$.

**Corollary 1.** *Consequently, $s_{u,v}$ is significant if $\dot{\mathcal{N}}_{\hat{G}}(u)$ and $\dot{\mathcal{N}}_G(v)$ have shared vertices, relative large edge weight product $\hat{e}_{u,i} e_{v,j}$, and vertex weight product $\hat{\lambda}_u \lambda_v$, which means $u$ and $v$ have similar **local structure**, particularly in the case that $u$ and $v$ are the same vertex.*

In conclusion, Equation (8) can be simplified as the summation of the joint vertices $\Phi = \hat{V} \cap V$ of the instance and category graph $G$ and $\hat{G}$. $\qquad \square$

## C  TRAINING ALGORITHM

The training algorithm of our proposed SchemaNet is shown in Algorithm 1 with initializing IR-Atlas and sparsification.

## D  EFFECTIVE RECEPTIVE FIELD OF VITS

In this section, we discuss the effective receptive field (ERF) of ViTs, which is crucial as the visual token in the intermediate layers of ViTs may relate to other positions due to MHSA, affecting the interpretability of the ingredients (visual word semantics). Although Luo et al. (2016) propose to measure the ERF for CNNs, it cannot be directly implemented to Transformer-base models. In this section, we propose a relatively simple yet effective approach to measuring the ERF of ViTs.

**Definition 3** (Transformer ERF). *Supposed that visual sequence $X \in \mathbb{R}^{n \times d}$ is fed into a ViT backbone with $N$ layers, $x_*$ is a randomly chosen anchor token in $X$, and $\varepsilon \in \mathbb{R}^d$ is the random vector drawn from Gaussian distribution $\mathcal{N}(0, I)$ normalizing to the unit vector. The Transformer ERF is the average Euclidean distance between the closest token $y$ to $x_*$ in the 2D token array and $x_*$ so that when $y$ is disturbed to $\hat{y} = y + \epsilon$ the output change of token $x_*$ is less than a predefined threshold $\delta_r > 0$.*

---

**Algorithm 1** SchemaNet optimizer with initialization and sparsification.

---

**Input:** $\mathcal{D} = \{(x_i, y_i)\}_{i=1}^{D}$: the training dataset with $D$ samples; $\hat{\mathcal{G}} = \{\hat{G}_c\}_{c=1}^{C}$: initial IR-Atlas; Backbone($\cdot$): the ViT backbone; Matcher($\cdot, \cdot$): the graph matcher; $\Omega$: the visual vocabulary; $\Theta$: the set of all trainable parameters; $\delta_t$: the sparsification threshold.

1: **procedure** INITIALIZATION($\mathcal{D}, \Omega$)
2:     Sample a subset $\tilde{\mathcal{D}} \subset \mathcal{D}$ as the probe dataset.
3:     $S_1, \ldots, S_C \leftarrow \varnothing, \hat{\mathcal{G}} \leftarrow \varnothing$
4:     **for** $(x, y) \in \tilde{\mathcal{D}}$ **do**
5:         $G \leftarrow$ FEAT2GRAPH(Backbone($x$), $\Omega$)
6:         $S_y \leftarrow S_y \cup G$
7:     **end for**
8:     **for** $c = 1, \ldots, C$ **do**
9:         $\hat{\mathcal{G}} \leftarrow$ AVERAGE($S_c$)                           $\triangleright$ compute average graph for each category
10:     **end for**
11:     **return** $\hat{\mathcal{G}}$
12: **end procedure**
13: **procedure** TRAINING($\hat{\mathcal{G}}, \mathcal{D}, \Omega$)
14:     **for** $(x, y) \in \mathcal{D}$ **do**
15:         **for** $\hat{e}_{i,j} \in \hat{E}$ **do**                      $\triangleright$ Removing redundant edges
16:             **if** $\hat{\lambda}_i < \delta_t$ or $\hat{\lambda}_j < \delta_t$ **then**
17:                 $\hat{e}_{i,j} \leftarrow$ NIL
18:             **end if**
19:         **end for**
20:         $G \leftarrow$ FEAT2GRAPH(Backbone($x$), $\Omega$)
21:         $\hat{y} \leftarrow$ Matcher($G, \hat{\mathcal{G}}$)
22:         Compute the final loss and gradient $\nabla_\Theta$ *w.r.t.* parameters $\Theta$.
23:         Update parameters $\Theta$ with AdamW optimizer.
24:     **end for**
25: **end procedure**
26: $\hat{\mathcal{G}} \leftarrow$ INITIALIZATION($\mathcal{D}, \Omega$)
27: TRAINING($\hat{\mathcal{G}}, \mathcal{D}, \Omega$)

---

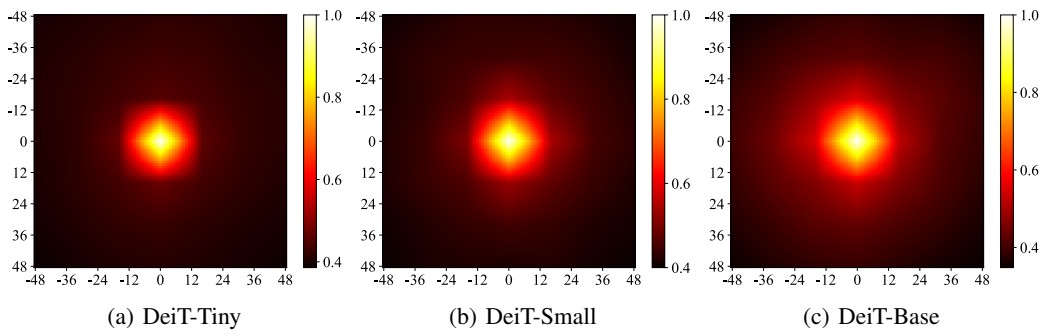

(a) DeiT-Tiny            (b) DeiT-Small            (c) DeiT-Base

Figure 7: The heat maps of the receptive field of an anchor token over its neighborhood region. Averaged visualization results of 64 random images of three ViTs (DeiT-Ti, DeiT-S, and DeiT-B) from Caltech-101 dataset are demonstrated. The full image size is $224 \times 224$, and we crop a small region with a size of $96 \times 96$ with the anchor token as the center for better visualization.

Figure 7 shows the visualization heatmap *w.r.t.* the change of the output anchor with different ViT backbones adopted in our method. As we can observe, for an input image size of $224 \times 224$, the output tokens with relatively significant changes are mainly distributed in the circle with a radius of 25. Therefore, in our method, all the ingredients are in charge of $50 \times 50$ patches corresponding to the input image.

# E EXPERIMENTAL DETAILS

## E.1 VISUAL VOCABULARY

Thanks to the relatively uniform representation of visual seman-
tics generated by DNNs, our visual vocabulary size $M$ can be
set to around $10 \times C$ with competitive performance. Table 2
shows the default choice of $M$ on different datasets. Results
with different vocabulary sizes are presented in Appendix G.
Furthermore, all the intermediate features are extracted from
the 9-th layer of the backbone ViTs, including DeiT-Tiny, DeiT-
Small, and DeiT-Base.

Table 2: Visual vocabulary size.

| Dataset | Size $M$ |
|---|---|
| CIFAR-10 | 128 |
| CIFAR-100 | 1024 |
| Caltech-101 | 1024 |

## E.2 IR-GRAPH

Before an instance-level graph $G$ and a category-level graph $\hat{G}$ are fed to the graph matcher, their
vertex weights and edge weights are normalized as follows. The vertex weights in $\Lambda$ are divided by
the sum of $\Lambda$

$$\Lambda_{\text{norm}} = \frac{\Lambda}{\sum_{i=1}^{|V|} \lambda_i}. \tag{23}$$

The adjacency matrix $E$ is divided by the row-wise summation:

$$E_{\text{norm}} = \begin{bmatrix} \frac{E_{1,:}}{\sum_{i=1}^{|V|} e_{1,i}} \\ \vdots \\ \frac{E_{|V|,:}}{\sum_{i=1}^{|V|} e_{|V|,i}} \end{bmatrix}. \tag{24}$$

Finally, the symmetric adjacency matrix $E_{\text{sym}}$ is defined as

$$E_{\text{sym}} = \frac{1}{2}(E_{\text{norm}} + E_{\text{norm}}^{\top}) \tag{25}$$

## E.3 GCN SETTINGS

Now we describe the detailed settings of the GCN module in the graph matcher. We adopt Lay-
erNorm (Ba et al., 2016) as $\text{Norm}(\cdot)$ function and rectified linear unit (ReLU) as the activation
in Equation (7). Besides, the embedding dimension $d_G$ is set to 256 for all the experiments.

## E.4 TRAINING DETAILS

The matcher and IR-Atlas in our method are optimized by AdamW (Loshchilov & Hutter, 2019) with
a learning rate of $10^{-3}$, weight decay of $5 \times 10^{-4}$, and cosine annealing as the learning rate decay
schedule. We implement our method with Pytorch (Paszke et al., 2019) and train all the settings for
50 epochs with the batch size of $64$ on one NVIDIA Tesla A100 GPU. All input images are resized
to $224 \times 224$ pixels before feeding to our SchemaNet. We adopt ResNet-style data augmentation
strategies: random-sized cropping and random horizontal flipping.

# F ADDITIONAL RESULTS

This section contains additional experimental results, highlighting the efficiency, robustness, and
extendability of our proposed schema inference.

## F.1 RESULTS ON IMAGENET

We further implement SchemaNet on mini-ImageNet (Vinyals et al., 2016) and ImageNet-1k (Deng
et al., 2009). We adopt DeiT-Small as the backbone, and the visual vocabulary size is set to 1024
for mini-ImageNet and 8000 for ImageNet-1k (due to the memory constraint). Particularly, as
implementing 1000 fully-connected category-level IR-Graphs is expensive, we keep at most 500

valid vertices for each class while all other vertices are pruned during the training process based on the initialized vertex weights. All other settings are identical to the experiments on Caltech-101. The results are presented in Table 3, drawing consistent conclusions as the main results in Table 1.

Table 3: Comparison results (top-1 accuracy) on ImageNet-1k and mini-ImageNet.

| Dataset | Base | Backbone-FC | BoVW-Deep | BagNet | SchemaNet | SchemaNet-Init |
|---------|------|-------------|-----------|--------|-----------|----------------|
| mini-ImageNet | 95.35 | 90.04 | 90.24 | 85.64 | 89.40 | **91.02** |
| ImageNet-1k | 79.90 | 69.23 | 58.95 | 68.92 | - | **74.05** |

## F.2 COMPARISON RESULTS OF USING ATTRIBUTION METHODS

Instead of using raw attention extracted from the backbone in Equation (3), we further employ feature attribution methods for computing the ingredient importance. Specifically, we compare the results using raw attention, CAM (Zhou et al., 2016), and Transformer-LRP (Chefer et al., 2021) in Table 4. We can observe that using raw attention is superior to the attribution methods in terms of accuracy and running time. Such results can be explained that: (1) the attribution methods compute a saliency map for each class (particularly, ViT-LRP conduct $C$-times backpropagation, consuming enormous time), which is significantly slower than our implementation; (2) further, as the backbone predicts soft probability distributions other than one-hot targets, the computed saliency maps for similar categories will be highlighted to some extent, impairing the graph matcher.

Table 4: Comparison results (top-1 accuracy and running time) of using different attribution methods on CIFAR-10 and CIFAR-100 datasets.

| Dataset | Raw Attention | | CAM | | ViT-LRP | |
|---------|------|------|------|------|------|------|
| | Acc | Time | Acc | Time | Acc | Time |
| CIFAR-10 | **95.96** | 0.8h | 93.58 | 3.6h | 91.28 | 53.8h |
| CIFAR-100 | **78.45** | 3.7h | 77.55 | 15.3h | - | - |

## F.3 ADVERSARIAL ATTACKS

To analyze the robustness of our proposed SchemaNet, we evaluate the pre-trained SchemaNet (from ImageNet-1k described in Appendix F.1) on two popular adversarial benchmarks: ImageNet-A (Djolonga et al., 2021) and ImageNet-R (Hendrycks et al., 2021). The comparison results are shown in Table 5, revealing that our method is more robust than the baselines.

Table 5: Adversarial attack results (top-1 accuracy) on ImageNet-A and ImageNet-R datasets.

| Dataset | Backbone-FC | BoVW-Deep | BagNet | SchemaNet-Init |
|---------|-------------|-----------|--------|----------------|
| ImageNet-A | 5.41 | 7.36 | 5.53 | **13.05** |
| ImageNet-R | 31.21 | 24.17 | 30.93 | **33.46** |

## F.4 EXTENDABILITY

We evaluate the extendability of our method by extending a trained SchemaNet to unseen tasks, while keeping the original framework frozen. As such, only the category-level IR-Graphs for the new tasks are optimized and inserted into the original IR-Atlas.

Specifically, the tasks are disjointly drawn from Caltech-101 dataset. The "Base" task has 21 classes, and task 1 to 4 has 20 classes. The backbone model, *i.e.*, DeiT-Tiny, is trained on the "Base" task and then is kept unchanged for the new tasks.

The experimental results are presented in Table 6, revealing limited performance degradation for incoming new tasks. Such results show that the graph matcher is only responsible for evaluating the graph similarity, while the category knowledge is stored in the imagination, *i.e.*, IR-Atlas.

Table 6: Top-1 accuracy results when extending SchemaNet to unseen tasks. The accuracies are evaluated on the corresponding task, and the average accuracy over all the tasks is given as well.

| | Average | Base | Task 1 | Task 2 | Task 3 | Task 4 |
|---|---|---|---|---|---|---|
| **Base** | 97.57 | 97.57 | | | | |
| **+Task 1** | 94.68 | 95.95 | 93.69 | | | |
| **+Task 2** | 93.36 | 95.14 | 93.06 | 92.11 | | |
| **+Task 3** | 93.63 | 95.55 | 93.06 | 92.47 | 93.75 | |
| **+Task 4** | 92.37 | 95.14 | 92.74 | 91.76 | 90.42 | 91.49 |

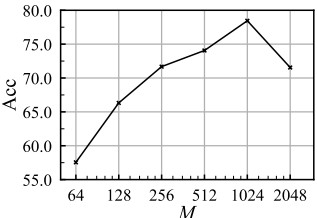

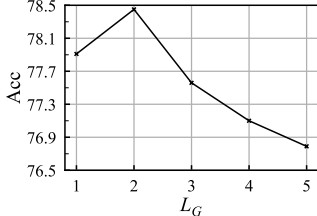

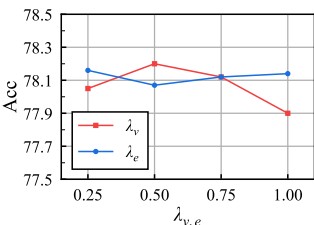

(a) SchemaNet accuracy with different visual vocabulary size $M$.

(b) SchemaNet accuracy when using GCN with various depth $L_G$.

(c) Sensitivity analysis of hyperparameters involved in Equation (11).

Figure 8: Ablation study and sensitivity analysis of hyperparameters of our proposed SchemaNet on CIFAR-100 backed with DeiT-Tiny.

## F.5 FEAT2GRAPH EVALUATION

We here present a performance bottleneck of our method, *i.e.*, the *Feat2Graph* module, in which a feature ingredient array is converted to IR-Graph without benefiting from GPU parallel acceleration. Specifically, as duplicated visual words may exist after the feature discretization, which happens from time to time, we implement this module in C++ to achieve better random access performance. The experiments are conducted on one NVIDIA Tesla A100 GPU platform with AMD EPYC 7742 64-Core Processor.

In Table 7, we show the average running time of each component with an input batch size of 64. We can see that without acceleration in parallel, the running time for "Feat2Edge" is significantly longer than others (about 1.75 ms per image). However, the inference time is still acceptable for real-time applications.

Table 7: Time costing (ms) of the components in SchemaNet with input batch size of 64.

| Dataset | Backbone | Feat2Vertex | Feat2Edge | Matcher |
|---|---|---|---|---|
| **CIFAR-10** | 4.89 | 7.00 | 19.8 | 1.95 |
| **CIFAR-100** | 4.31 | 9.50 | 101.7 | 1.30 |
| **Caltech-101** | 4.12 | 8.12 | 111.7 | 1.42 |

## G ABLATION STUDY

### G.1 VISUAL VOCABULARY SIZE

Figure 8(a) shows SchemaNet accuracy when using different sizes of visual vocabulary. We can observe that even with a relatively small size, *e.g.*, $M = 256$ on CIFAR-100, the performance is still competitive (about 7% absolute degradation). Besides, for the case of $M = 2048$, however, the performance suffers from low generalizability of the vocabulary.

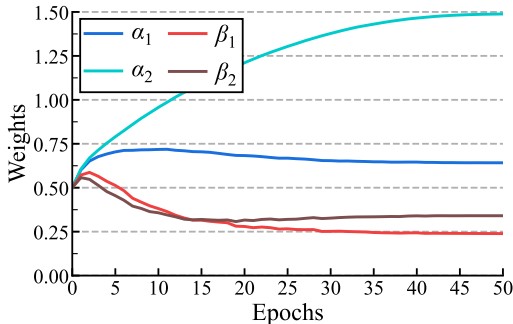

Figure 9: Learning curves of $\alpha_{1,2}$ and $\beta_{1,2}$ with DeiT-Tiny backbone on Caltech-101 dataset.

## G.2 NUMBER OF GRAPHCONV LAYERS

The effect of using different depths of GCN is illustrated in Figure 8(b). As $L_G$ increases to 5, the accuracy on CIFAR-100 decreases rapidly, yielding that deep GCN matcher impairs the graph matching performance, which has been explored due to the over-smoothing caused by many convolutional layers (Zhao & Akoglu, 2020; Li et al., 2018). In our method, however, the graph matcher with only two layers of GraphConv is proven to be adequate for high performance and interpretability.

## G.3 ABLATION OF WEIGHT COMPONENTS

The effect of $\lambda^{\text{CLS}}$ and $\lambda^{\text{bag}}$ defined in Equation (3), and $e^{\text{attn}}$ and $e^{\text{adj}}$ defined in Equation (6) are shown in Table 8. By setting the corresponding weight ($\alpha_{1,2}$ and $\beta_{1,2}$) to zero once at a time, we are able to analyze the components individually. In general, removing any term will lead to varying degrees of performance degradation. More significantly, when removing the term $\lambda^{\text{bag}}$, the accuracy decreases by 6.03% on CIFAR-100, revealing that the statistics of the ingredients help filter noisy vertices.

Table 8: Ablation study of the components defining the vertex and edge weights with DeiT-Tiny as the backbone DNN.

| Settings | Accuracy | |
| --- | --- | --- |
| | CIFAR-10 | CIFAR-100 |
| Learnable $\alpha_{1,2}, \beta_{1,2}$ | **95.96** | **78.20** |
| Fixed $\alpha_1 = 0, \alpha_2 = 1$ | 95.88 | 75.91 |
| Fixed $\alpha_1 = 1, \alpha_2 = 0$ | 95.68 | 72.17 |
| Fixed $\beta_1 = 0, \beta_2 = 1$ | 95.85 | 77.31 |
| Fixed $\beta_1 = 1, \beta_2 = 0$ | 95.89 | 77.84 |
| Fixed $\alpha_{1,2} = \beta_{1,2} = 0.5$ | 95.92 | 75.62 |

The learning curves of the learnable weights are shown in Figure 9, from which we can observe that the model tend to adopt a relatively larger $\alpha_2$ (for visual word count) while keep $\lambda_v^{\text{CLS}}$ term for filtering noisy and background ingredients.

## H SENSITIVITY ANALYSIS OF HYPERPARAMETERS

## H.1 SPARSIFICATION THRESHOLD

Table 9 presents the SchemaNet-Init top-1 accuracy trained with different sparsification threshold $\delta_t$. We can observe that an appropriate value of $\delta_t$, *e.g.*, 0.01, will boost the model performance.

Table 9: Sensitivity analysis of sparsification threshold $\delta_t$.

| $\delta_t$ | 0 | 0.001 | 0.01 | 0.02 | 0.05 | 0.1 |
| --- | --- | --- | --- | --- | --- | --- |
| Acc | 77.27 | 77.96 | 78.45 | 78.43 | 77.51 | 76.92 |

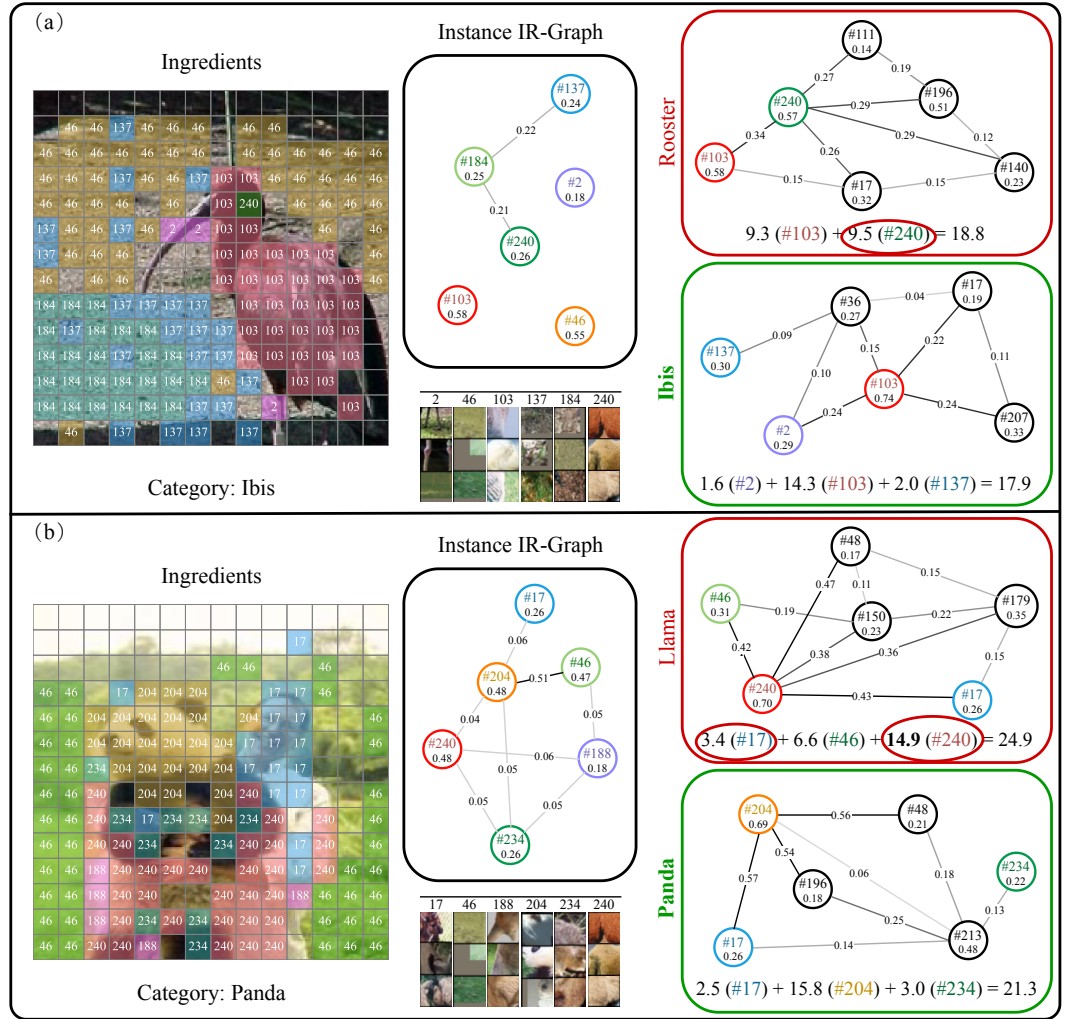

Figure 10: Illustration of misclassified samples. Better view in color.

## H.2 Loss Weights

Figure 8(c) shows the sensitivity analysis of $\lambda_v$ and $\lambda_e$ in Equation (11) that constrains the complexity of IR-Atlas. We evaluate the top-1 accuracy performance on the CIFAR-100 dataset with DeiT-Tiny as the backbone. The curve of $\lambda_v$ shows that neither dense nor extreme sparse IR-Atlas impairs the performance, while our method is more robust when changing $\lambda_e$.

## I More Visualizations

### I.1 Visualization and Analysis of Misclassified Examples

Figure 10 shows two misclassified examples along with the classification evidence. In Figure 10(a), the object is misclassified to a fine-grained category "rooster" because of a noise ingredient #240, which should be #103. Unfortunately, ingredient #240 is a crucial vertex in the rooster's IR-Graph, contributing about 9.5 absolute gains in the logit, leading to misclassification. Figure 10(b) shows a more complicated example. We can observe that instead of discretizing the appeared human face to the "face" ingredients, the backbone provides features closer to #17, which is more similar to the animal's face. Moreover, some part of the object's body is assigned to #240 rather than the panda's body (#204). Thus, it creates a remarkable pattern, *i.e.*, the interaction between #17 and #240, which

is the crucial local structure in the llama's graph. As a result, #240 and its local structure contribute about 14.9 gains in the logit. Although our schema inference framework is capable of revealing why an image is misclassified by highlighting the key points, in future work, we must explore a more compatible feature extractor that can generate more robust local features.

## I.2    VISUALIZATION OF INGREDIENTS

In Figure 11, we visualize the whole set of ingredients on Caltech-101 for visualizations in Figure 5 and Figures 12 to 14. We extract 256 ingredients on Caltech-101 for a better view. For each cluster center generated from $k$-means clustering, we select the top-40 tokens that are closest to it and show the corresponding image patch ($50 \times 50$ in pixel, delineated in Appendix D). We can observe that image patches of the same cluster share the same semantics.

## I.3    VISUALIZATION OF IR-ATLAS AND INSTANCE IR-GRAPHS

We provide more visualization examples on Caltech-101 dataset, shown in Figures 12 to 14.

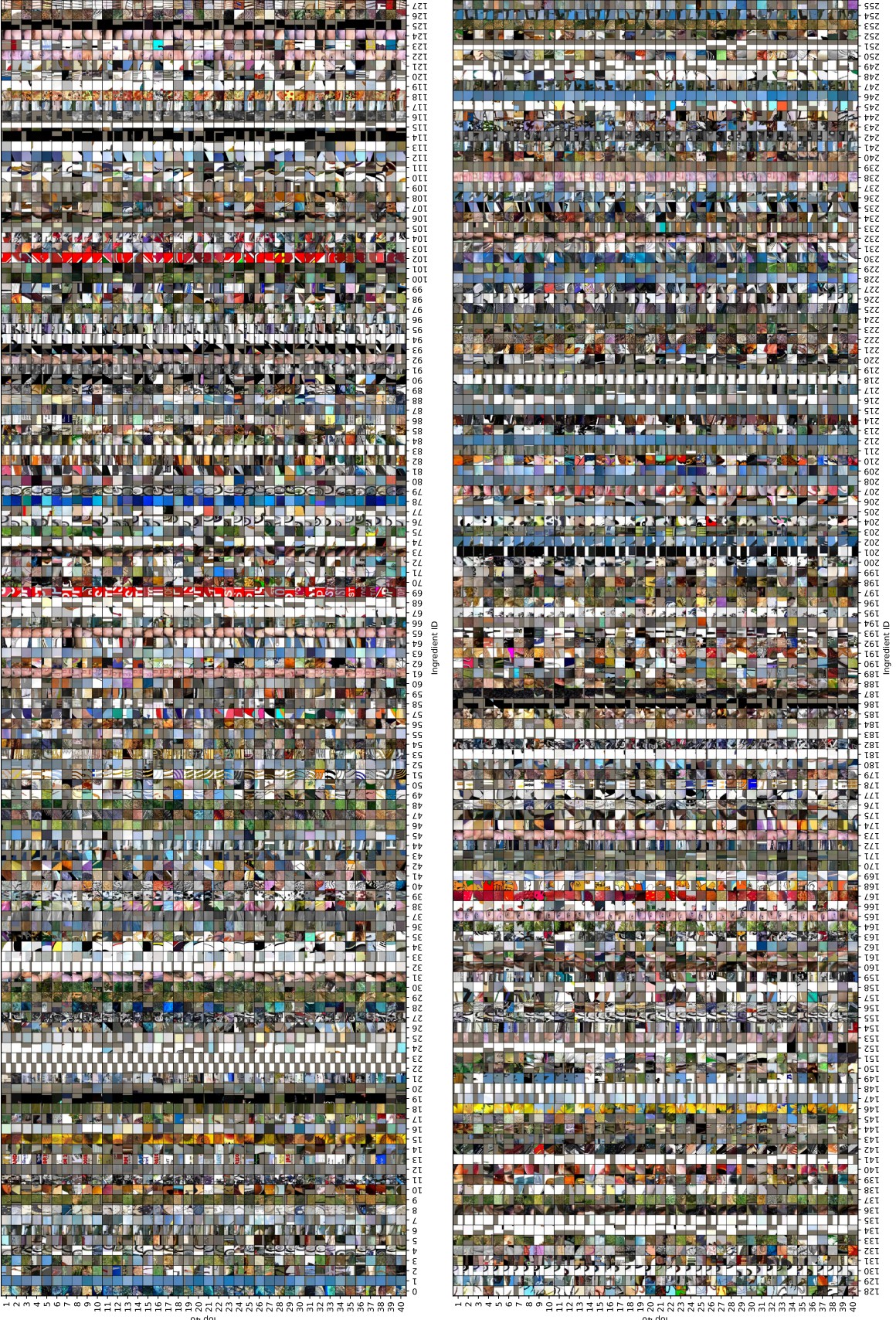

Figure 11: Ingredient visualization on Caltech-101 dataset. Please zoom in for a better view.

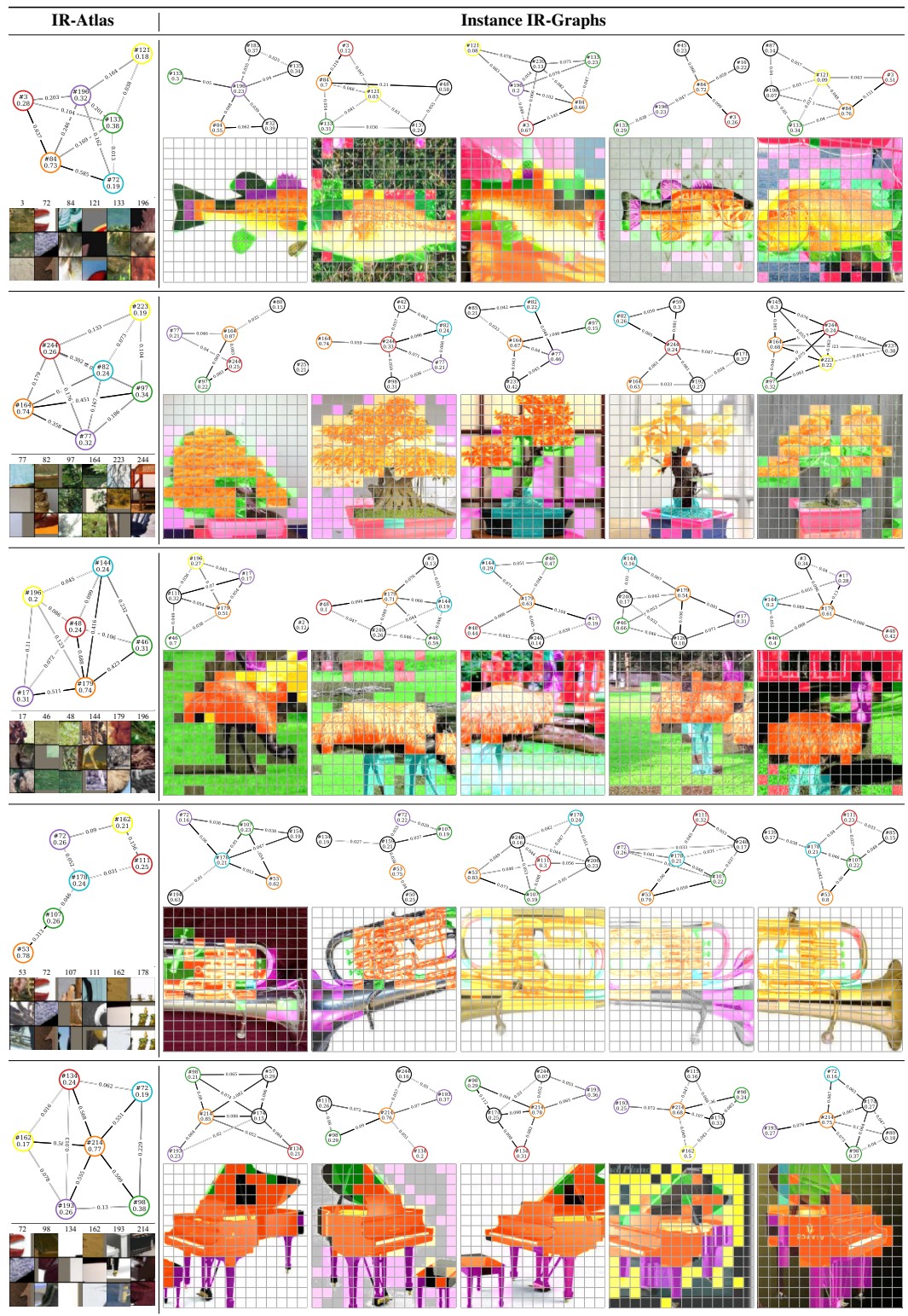

Figure 12: Examples of the learned IR-Atlas and instance IR-Graphs randomly sampled from five categories ("bass", "bonsai", "emu", "euphonium", and "grand piano") on Caltech-101.

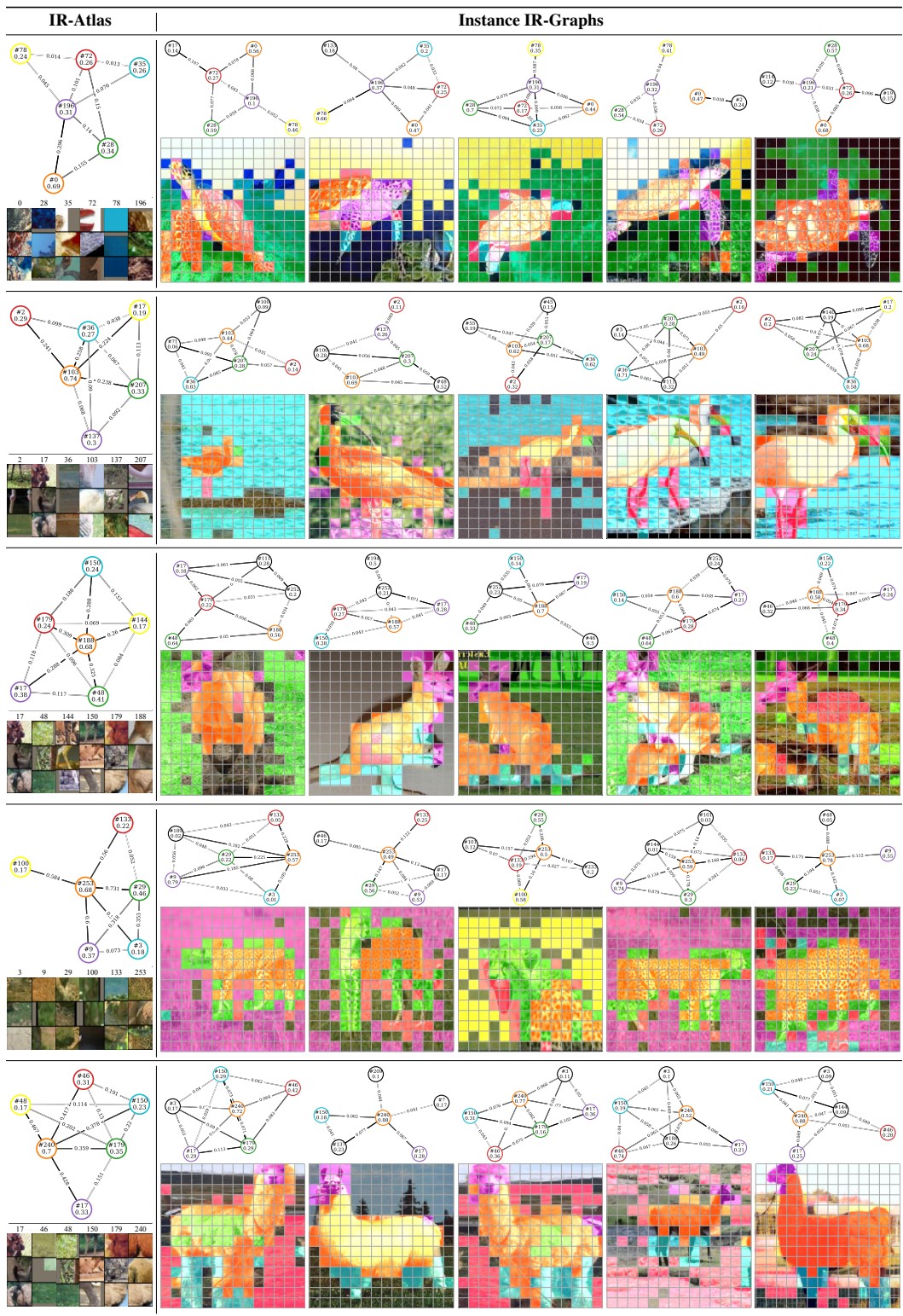

Figure 13: Examples of the learned IR-Atlas and instance IR-Graphs randomly sampled from five categories ("hawksbill", "ibis", "kangaroo", "leopards", and "llama") on Caltech-101.

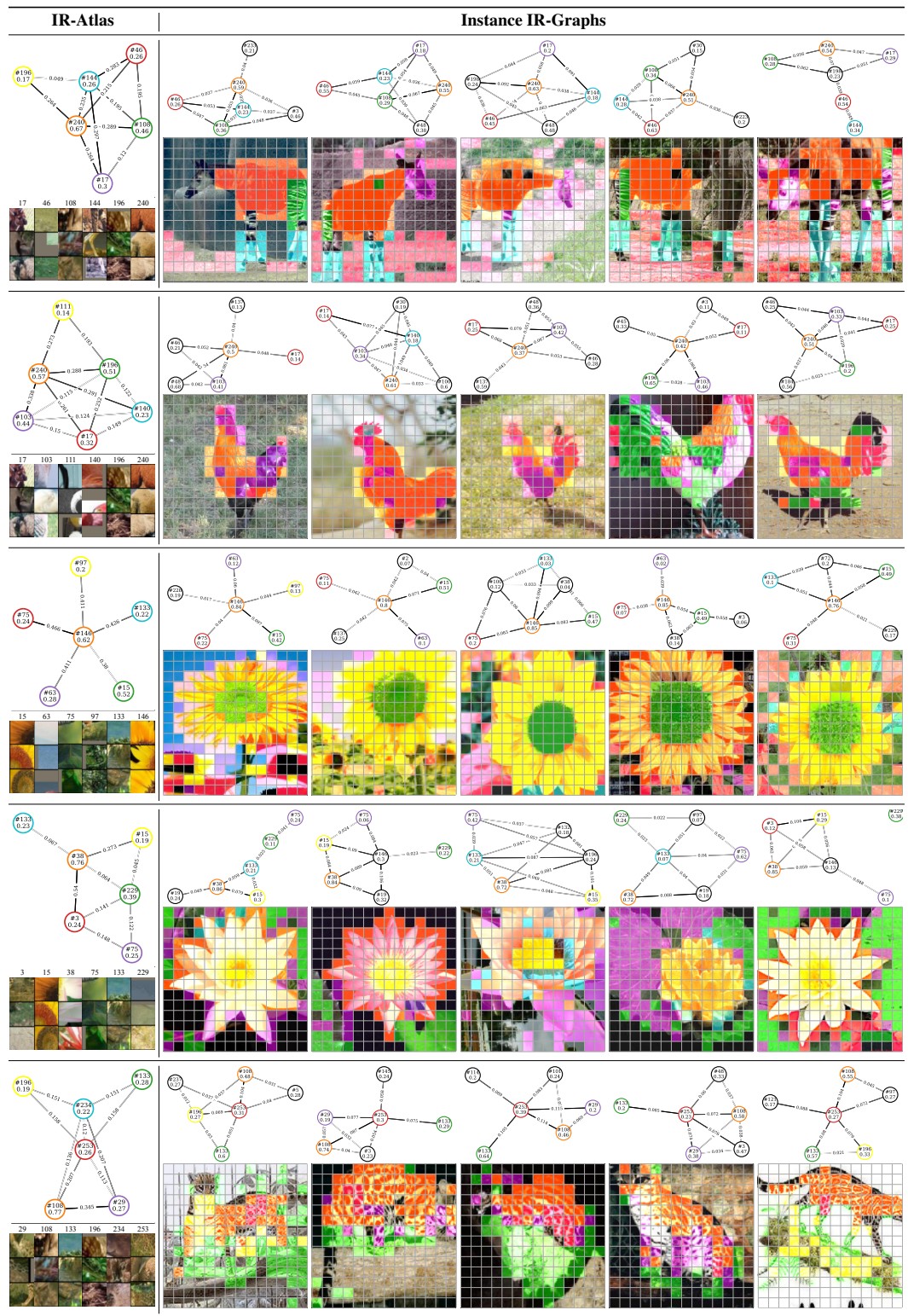

Figure 14: Examples of the learned IR-Atlas and instance IR-Graphs randomly sampled from five categories ("okapi", "rooster", "sunflower", "water lilly", and "wild cat") on Caltech-101.

