# OpenReview forum: "Schema Inference for Interpretable Image Classification"
_ICLR.cc/2023/Conference — ICLR 2023 poster_

### Official Review · Reviewer_S8GX · 2022-10-22

**Confidence:** 5
**Correctness:** 3
**Technical Novelty And Significance:** 3
**Empirical Novelty And Significance:** 3
**Recommendation:** 8

**Clarity, Quality, Novelty And Reproducibility:**


The quality and clarity are median. The originality is good.


**Details Of Ethics Concerns:**


There is no ethics concern.


**Strength And Weaknesses:**



(Positive) The authors think their method is an analogy with the human reasoning mechanism via impression matching. Such motives are very reasonable. I like it.

(Negative) The conceptual clustering shown in the lower left corner of Figure 2 is very interesting. It is the basis for the soundness of the method proposed in this article. I hope the authors give more examples (if possible, show all of them), especially in large resolutions.

(Positive) I am happy that, instead of relying on deep features, the desired outputs from schema rely on visual word relationships.

(Positive) Thanks to the authors, Figures 1 and 2 are very clear and easy to understand.

(Negative) In my opinion, the authors should validate the effectiveness of their method on more standard data such as ImageNet.

(Negative) Authors should compare their machine-learned knowledge with human knowledge and make it clear whether their machine-learned knowledge is more advanced than human knowledge. Is it worthwhile for humans to learn this independent knowledge from these machines?

(Negative) The method proposed in this paper is an overly complex engineering system, and there are many factors to be analyzed. The definition of Feat2Vertex (Equation (3) is necessary for workability but not elegant. There should be a tradeoff between the similarity at the semantic level as well as the adjacency relationship at the spatial level in Feat2Edge, which is also not elegant.

(Negative) The authors assign a dG dimensional random vector x ∈ R 174 dG to each vertex before feeding the graph to the GCN. Does this mean that there is randomness in every inference? What do you think of this randomness?

(Negative) In order to avoid the high complexity of fully connected graphs for IR-Atlas, this paper averages each instance graph. This involves engineering. Please analyze the impact of hyperparameters such as thresholds.



**Summary Of The Paper:**


Summary: This paper proposes a new machine model from the perspective of human cognition. This is very good. However, the system in this paper is too complex and is not concise and elegant enough. Its usefulness (especially in real environments or large datasets) remains to be further verified.


**Summary Of The Review:**



See "Summary Of The Paper." I think this paper is a borderline paper, leaning toward being accepted.

---

> ### Author Response · Authors · 2022-11-19
> **To Reviewer S8GX (2/2)**
>
>
> **Q4:** The method proposed in this paper is an overly complex engineering system, and there are many factors to be analyzed. The definition of Feat2Vertex (Equation (3) is necessary for workability but not elegant. There should be a tradeoff between the similarity at the semantic level as well as the adjacency relationship at the spatial level in Feat2Edge, which is also not elegant.
>
> **A4:** Thanks for the constructive suggestions. We want to clarify that, to our best knowledge, we are the first attempt to utilize the relationship between visual words to build relation graphs for interpretable predictions. We agree with the reviewer that the method designing can be improved, and we will improve our method in a more concise and practical form in our future work.
>
>
> **Q5:** The authors assign a $d_G$ dimensional random vector $x\in R^{d_G}$ to each vertex before feeding the graph to the GCN. Does this mean that there is randomness in every inference? What do you think of this randomness?
>
>
> **A5:** We apologize for the confusion. We have revised Section 2.3 to avoid misunderstanding.
> In fact, the random vector assigned to each vertex as the vertex embedding is only for initialization (for making them orthogonal with each other in Theorem 1).
> During the training procedure, such the property still holds, as shown in Figure 6.
> In the evaluation stage, the vertex embeddings are fixed. So, there is no randomness in every inference.
>
>
> **Q6:** In order to avoid the high complexity of fully connected graphs for IR-Atlas, this paper averages each instance graph. This involves engineering. Please analyze the impact of hyperparameters such as thresholds.
>
>
> **A6:** Thanks for your comments. We have updated our paper and added more sensitivity analysis of the hyperparameters in Appendix H. The results of the SchemaNet-Init top-1 accuracy trained with different sparsification threshold $\delta_t$ are listed below. We can observe that an appropriate value of $\delta_t$ (e.g., 0.01) will boost the model performance.
>
> | $\delta_t$ | 0 | 0.001 | 0.01 | 0.02 | 0.05 | 0.1 |
> |-|-|-|-|-|-|-|
> | Acc | 77.27 | 77.96 | 78.45 | 78.43 | 77.51 | 76.92 |
>
>
> Thank you again for your efforts in reviewing our paper. We hope our response can address your concerns.

---

> ### Author Response · Authors · 2022-11-19
> **To Reviewer S8GX (1/2)**
>
> We appreciate the reviewer for insightful feedback to help improve our work. We have also made some changes to our paper, and we will address the concerns and questions for the reviewer as follows. Please let us know if you have other questions. We are happy to follow up on additional feedback.
>
>
> **Q1:** The conceptual clustering shown in the lower left corner of Figure 2 is very interesting. It is the basis for the soundness of the method proposed in this article. I hope the authors give more examples (if possible, show all of them), especially in large resolutions.
>
> **A1:** Thanks for your helpful comment. Indeed, the visual words in our paper relate to the most representative semantics in the probe data, providing the intuitive interpretability of our method. We have updated the manuscript to show all the visual words in Figure 10 in Appendix I, which are extracted on Caltech-101. It can be observed that the visual words generally represent the common visual patterns in the foreground and the background, such as the human eyes (visual word #122) and the green grass (visual word #49).
>
> **Q2:** In my opinion, the authors should validate the effectiveness of their method on more standard data such as ImageNet.
>
> **A2:** Thanks for the reviewer's insightful suggestions. We update our paper and provide additional results on ImageNet in Appendix F.1. We adopt DeiT-Small as the backbone, and the visual vocabulary size is set to 1024 for mini-ImageNet and 8000 for ImageNet-1k (due to the memory constraint). Notably, as implementing 1000 fully-connected category-level IR-Graphs is expensive, we have tried our best to improve the SchemaNet so that it supports the dynamic sparsity of the IR-Atlas. As shown in Table 5, the results are consistent with the results in Table 2 in the main text: SchemaNet also achieves superior performance than the baseline methods, validating the effectiveness of our method.
>
>
> **Q3:** Authors should compare their machine-learned knowledge with human knowledge and make it clear whether their machine-learned knowledge is more advanced than human knowledge. Is it worthwhile for humans to learn this independent knowledge from these machines?
>
> **A3:** Thanks for pointing this out.
> We have added more discussions about the learned IR-Atlas in Section 3.4.
> Briefly, the learned IR-Atlas is somewhat similar to human intuition from the three aspects:
> (1) The visualization of the ingredients shows that they can represent explicit semantics, which is easy for humans to understand.
> Besides, the learned vertex weights are also consistent with human knowledge as the object ingredients get higher weights than the background ingredients.
> (2) The learned edge weights tend to connect foreground object parts with their surroundings while ignoring the background connections, helping distinguish fine-grain categories.
> (3) As illustrated in Figure 5 and 11 to 13, the learned category graphs are able to capture the general character of instances, forming the abstract imaginations of the categories.
> Finally, we agree with the reviewer that more studies on human measurements should be conducted. We will compare machine-learned and human knowledge human and introduce their combination in our future work.

---

> ### Comment · Reviewer_S8GX · 2022-12-11
> **Responses to the authors' responses:**
>
>
> I am delighted that the authors have responded well to all my concerns, especially since they reported the results on ImageNet in the update. The authors also responded well to my other concerns, including suggestions for more visualizations, discussions of alignment with humans, randomization confusion, and engineering hyperparameter analysis.
>
> I also read the excellent and valuable comments and suggestions from other reviewers. Congratulations to the authors for being recognized.
>
> Finally, I will raise my rating.

---

### Official Review · Reviewer_zHVw · 2022-10-24

**Confidence:** 4
**Correctness:** 3
**Technical Novelty And Significance:** 3
**Empirical Novelty And Significance:** 3
**Recommendation:** 6

**Clarity, Quality, Novelty And Reproducibility:**

Clarity:
The paper is overall clear. However, there are some parts that can be improved as detailed below.

First, terminologies are not so clear. The authors invent quite some new terminology such as "feature ingredients", "imagination of all categories", which are very unconventional words in the machine learning literature. This brings some difficulties to follow the description about the method. It is suggested to replace these words or explain why using them.

Second, the concept of interpretability is not clearly analyzed or illustrated. For instance, in Figure 1 and Figure 5, the authors use quite some different colours to illustrate the learned interpretability. However, it is not clearly explained what different colours means and how the model provide good interpretability.

Quality & Novelty.
Although the proposed method brings some technical novelty to existing XAI literature. The evaluation of the proposed method is weak.

First, evaluation compared to related representative methods [a,b] are not given.

Second, modern evaluation metrics (such as perturbation test) used to evaluate the model interpretability is not used, please refer to [c] for more context.

Third, the visualization about model interpretability is not clearly explained.

Overall, the insufficiency in evaluation makes the overall technical novelty weak.

[a] Learning Deep Features for Discriminative Localization, CVPR2016
[b] Attention Branch Network: Learning of Attention Mechanism for Visual Explanation, CVPR2019
[c] Transformer Interpretability Beyond Attention Visualization, CVPR2021

Reproducibility:
The authors provide some code to reproduce their method.


**Strength And Weaknesses:**

+ The proposed method of learning graphs to make the deep neural network more interpretable is interesting and novel. The idea makes a difference as compared to existing works in XAI.

- Although the proposed method brings some new aspects to XAI, the comparison in experiments is weak. The authors mainly compare the proposed approach to a couple of relatively simple methods such as the bag-of-word representations. However, there are quite some other existing representative methods such as class activation map (CAM) [a], and ABN [b], and [c] for vision transformer. Some of these existing methods [a,b] also contain learnable interpretability modules in the deep neural networks and provides interpretable predictions for image classification.

- The evaluation and analysis of the interpretability is also relatively unclear and weak. Most of the experiments are conducted to show the accuracy on classification. However, there is not much analysis about the model interpretability. The only visualization in Figure 5 to shown the interpretability is unclear. For instance, what does the different colours refer to? How does these interpretability compared to human interpretability? How does these interpretability contribute to the model predictions? It is suggest to follow existing literature in XAI [c] to give both quantitative and qualitative evaluation about the interpretability.

[a] Learning Deep Features for Discriminative Localization, CVPR2016
[b] Attention Branch Network: Learning of Attention Mechanism for Visual Explanation, CVPR2019
[c] Transformer Interpretability Beyond Attention Visualization, CVPR2021

**Summary Of The Paper:**

This paper studies interpretability for image classification. The authors propose a schema-based architecture (SchemaNet) based on vision transformers, which first extract features from pre-trained backbone and form a so-called ingredient relation graph (IR-Graph) with a feature to graph (Feat2Graph) module. The IR-Graph is matched with a set of imaginations (so-called IR-Atlas) with a matching network for making predictions. To test the SchemaNet, the authors conduct experiments on CIFAR10/100, and Caltech101, in comparison to the baseline without SchemaNet, bag-of-visual-words (BoVM) representation, BagNet, and show that SchemaNet performs better or on par with the baseline. Further, the authors also show some visualization of the learned IR-Graph and IR-Atlas.

**Summary Of The Review:**

The paper studies interpretabiltiy for image classification and propose a new method with learnable graphs. Experiments are given to showcase is classification performance in CIFAR and Caltech. However, the comparison to existing related methods is not comprehensive. The analysis and evaluation about the model interpretabiltiy is also relatively unclear and weak as compared to existing works.

---

> ### Author Response · Authors · 2022-11-19
> **To Reviewer zHVw (2/2)**
>
>
> **Q3:** How does these interpretability compared to human interpretability? How does these interpretability contribute to the model predictions?
>
> **A3:** Sorry for the confusion. We have revised Section 3.4 to give a detailed description of the interpretability.
> In summary,
> (1) the visual words related to the graph's vertices represent the dataset's specific semantics. Besides, the learned weights of each vertex are also consistent with human intuition as the object ingredients get higher weights than the background ingredients.
> (2) The learned edge weights highlight the connections between the foreground object parts and the surroundings, consistent with human intuition.
> (3) Driven by sufficient instant graphs, the class graphs show the capacity to capture and model the general character of instances and eventually form the abstract imaginations of the categories.
>
> Further, as described in Theorem 1 and Corollary 1, the graph similarity between the instance and category graphs is dominated by the joint vertices that occur in both graphs, with two components: vertex importance and the similarity of the local structure of each critical vertex. Moreover, the visualizations in Section 3.4 and Appendix I support the interpretability.
>
>
> **Q4:** Some terminologies are not so clear. The authors invent quite some new terminology such as "feature ingredients", "imagination of all categories", which are very unconventional words in the machine learning literature.
>
> **A4:** We thank the reviewer for the suggestion on terminologies, and we would like to give more discussions on the two terminologies. For "feature ingredients", we provide a high-level definition in the Introduction (line 45) and a strict definition in Section 2.2 (line 143), which are, in fact, the indices of the visual words.
> The purpose of using "feature ingredients" is to emphasize that only indices are adopted to generate the IR-Graph rather than deep features, as one may refer to the visual word as the embedding vector.
>
> For "imagination of all categories", we also give a definition in the Introduction (line 41).
> As we are inspired by Kant's philosophy to use SchemaNet for associating specific instances to abstract imaginations, the imagination of all categories is realized as the category-level IR-Atlas.
>
>
> Thank you again for your efforts in reviewing our paper. We are looking forward to your feedback. Please let us know if you have other questions. We are happy to further answer the questions.
>
> **References**
>
> [1] Learning Deep Features for Discriminative Localization, CVPR2016
>
> [2] Attention Branch Network: Learning of Attention Mechanism for Visual Explanation, CVPR2019
>
> [3] Transformer Interpretability Beyond Attention Visualization, CVPR2021

---

> ### Author Response · Authors · 2022-11-19
> **To Reviewer zHVw (1/2)**
>
> We truly appreciate the reviewer's constructive suggestions and comments on our work, and we will address the concerns and questions for the reviewer as follows.
>
> **Q1:** The comparison in experiments is weak. The authors should compare this work with some novel XAI methods [1-3].
>
> **A1:** Thanks for the insightful question and suggestion.
> We have conducted more comparison experiments and updated our manuscript to make more comparisons between our method and novel XAI methods.
> First, we add discussions between our method and novel feature attribution methods for XAI (including [1-3]) in the related works in Appendix A.4.
> Next, we further compare our method (using raw attention) with CAM [1], and ViT-LRP [3] in Appendix F.2. Results in Table 4 show that our method can achieve better performance and less running time compared to CAM and ViT-LRP. We contribute these results as (1) the attribution methods compute a saliency map for each class (particularly, ViT-LRP conducts C-times backpropagation, consuming enormous time), which is significantly slower than our implementation; (2) further, as the backbone predicts soft probability distributions other than one-hot targets, the computed saliency maps for similar categories will be highlighted to some extent, impairing the graph matcher.
>
> Here, we also want to clarify why we don't perform comparison with ABN [2]: for one reason ABN is designed for the CNN architectures, which is not suitable for transformer backbones in our work; and for another the transformer in our method is discretized by the visual words and are incapable of being learned using ABN method. As such, we don't compare ABN with SchemaNet.
>
>
> **Q2:** The evaluation and analysis of the interpretability is also relatively unclear and weak. What does the different colours refer to? It is suggest to follow existing literature in XAI [3] to give both quantitative and qualitative evaluation about the interpretability.
>
> **A2:** Thank you for your constructive suggestion. Different colors in Figure 5 represent different visual words. For example, in the first row, the visual word #204 (the orange circle) refers to the white fur of the giant panda, and in the example images, the visual word #204 is mapped to the face of pandas (i.e., the orange image patches).
>
> We have also revised our paper and added the quantitative and qualitative evaluation of the interpretability in Section 3.3 using the positive and negative perturbation tests following ViT-LRP [3]. For a fair comparison, the attention values to the CLS token $\phi^{CLS}$ are extracted as the pixel relevance for BoVW-Deep, BagNet, and SchemaNet-Init with DeiT-Tiny. As shown in Figure 4, the positive perturbation test of our method behaves similarly to BoVW while significantly outperforming BagNet.
> In the negative perturbation test, our method achieves better performance by a large margin (the area-under-the-curve of ours is about 5.71% higher than BagNet on CIFAR-10 and 11.23% higher than BagNet on CIFAR-100). In summary, our SchemaNet has achieved superior performance on the quantitative and qualitative evaluation of interpretability compared to other methods.

---

> > ### Comment · Reviewer_zHVw · 2022-12-03
> > **Updated comments**
> >
> > Thank you to the authors for the rebuttal and the new experiments. Most of my concerns have been addressed. I changed my recommendation score to 6.
> >
> > However, in the final version, I would suggest to add the comparison of visualization to the other existing XAI methods (CAM, ViT-LRP), which can help highlighting the uniqueness of this method. Other approaches (CAM, ViT-LRP) mostly give attention based explanations, but this paper presents a new type of explanation. I find this part is novel but not sufficiently highlighted, and discussed.

---

> > > ### Author Response · Authors · 2023-02-22
> > > **Response to the updated comments**
> > >
> > > Thanks for the nice suggestion! Post-hoc XAI methods like CAM mainly explain a given deep model by producing saliency maps as visualization, highlighting which parts of the image contribute more to the model prediction. On the other hand, our method is a new type of intrinsic interpretable method, which by design, is able to produce explanations of how the predictions are made as the graph-matching visualizations in Figures 5 and 12 to 14. However, our method cannot generate saliency maps like CAM because the deep features do not participate in the decision-making process (only the semantics and their relations are preserved in IR-Graphs). Nevertheless, CAM or other XAI methods can be adopted in our method, particularly in Feat2Graph, and we report the experimental results in Section F.2. Finally, thank you again for the valuable and generous suggestions for improving the quality of our paper!

---

> ### Author Response · Authors · 2022-11-29
> **Looking forward to your further feedback**
>
> Dear reviewer zHVw,
>
> We sincerely thank you for the valuable suggestions that have helped us improve the quality of the paper significantly.
>
> We hope our responses to the experimental comparisons and the evaluation and analysis of the interpretability could address your concerns. And we would appreciate the opportunity to discuss this further if our responses have not already addressed your concerns.
>
> Yours sincerely,
>
> Authors

---

### Official Review · Reviewer_bkvM · 2022-10-24

**Confidence:** 3
**Correctness:** 3
**Technical Novelty And Significance:** 3
**Empirical Novelty And Significance:** 3
**Recommendation:** 6

**Clarity, Quality, Novelty And Reproducibility:**

This work is clearly written, is of a reasonable quality, and is novel. I believe that the work has no issue with reproducibility.

**Details Of Ethics Concerns:**

There are no ethical concerns.

**Strength And Weaknesses:**

Strengths:

- Novelty: The proposed SchemaNet is novel -- the idea of turning image classification into a graph matching problem has not been explored before.
- Clarity: The paper is in general clearly written.

Weaknesses:

- Motivation: I suppose that the motivation for turning image classification into a graph matching problem is to build a model with a more interpretable reasoning process. However, the paper did not discuss how interpretable the visual words and the class IR-Graphs are. In particular, what are the meanings of the visual words? Do the learned class IR-Graphs make sense? The paper would be significantly better, if the authors can include interpretations of some class IR-Graphs. Without such interpretations, it would be difficult to see why we want to turn an image classification problem into a graph matching problem.
- A related issue: Since the authors used a graph convolutional neural network (GCN) to match an instance IR-Graph with each of the class IR-Graphs, and a GCN is not interpretable in general, how would the authors explain why an instance IR-Graph (representing a particular input image) is similar to a class IR-Graph? Again, without interpretability, I fail to see why we want to turn an image classification problem into a graph matching problem.

**Summary Of The Paper:**

In this paper, the authors proposed a new network architecture, called a SchemaNet, that turns an image classification problem into a graph matching problem. In particular, a SchemaNet consists of: (1) a pre-trained visual transformer backbone, that extracts image features from an input image; (2) a Feat2Graph module, that turns deep image features of an image into an instance IR-Graph, where vertices represent concepts and edges represent relations between concepts; (3) a set of IR-Graphs (called "IR-Atlas"), each of which represents prototypical concepts and their relations for each class; and (4) a graph neural network that computes the similarity between the instance IR-Graph and each of the class IR-Graphs in the IR-Atlas. The final prediction is based on the latter similarity -- the class whose IR-Graph has the highest similarity with the instance IR-Graph is used as the predicted class of the input. The authors compared their SchemaNet with a number of baseline models, including the base transformer, bag-of-visual-words over deep image features, and BagNet, on CIFAR-10, CIFAR-100, and Caltech-101, and found that their SchemaNet achieved competitive classification accuracy as the baseline models.

**Summary Of The Review:**

Based on the strengths and the weaknesses discussed above, I believe that this work has value, but needs improvement in explaining how to interpret the learned class IR-Graphs and how an instance IR-Graph is similar to a particular class IR-Graph. Without such interpretability, it is difficult to see the motivation of this work.

---

> ### Author Response · Authors · 2022-11-19
> **To Reviewer bkvM**
>
> We thank the reviewer for the valuable comments for improving our paper, and we will address the concerns and questions as follows.
>
> **Q1:** What are the meanings of the visual words?
>
> **A1:** Thanks for the helpful comments, and we apologize for the misleading. We have revised the manuscript to make more discussions on the interpretability of the visual words and the class IR-Graphs in Section 3.4.
> Similar to the visual words adopted by BoVW [1], our visual words are obtained via k-mean clustering on deep feature vectors, representing the most common semantics easily understood by humans.
> We further demonstrate the whole set of visual words on Caltech-101 in Figure 11.
> As shown in Figure 5 in the main text, in each row, we visualize five representative visual words of each learned IR-Atlas and some instance IR-Graphs, where each visual word can be consistently mapped to some object parts in input images by calculating and finding the closet distances between deep features. Here, we also relate each visual word to a circle with specific colors. Take the first row, for example, it can be observed that visual word #48 represents the green grass, and visual word #204 represents the white fur of the giant panda in the IR-Atlas. In the example Instance IR-Graphs, the two visual words are mapped separately to the background (the blue patches) and the face of the pandas (the orange patches). In conclusion, the visual words in our method represent typical information of the given probe data and act as the vertices to help us build the interpretable IR-Graphs with their semantic-rich characteristic.
>
>
> **Q2:** Do the learned class IR-Graphs make sense? The paper would be significantly better, if the authors can include interpretations of some class IR-Graphs.
>
> **A2:** Thanks for the suggestions. We have included more discussions of the class IR-Graphs in the updated manuscript in Section 3.4.
> The interpretability of the category-level IR-Atlas can be summarized as
> (1) the visual words related to the vertices in the graph represent specific semantics of the dataset. Besides, the learned weights of each vertex are also consistent with human intuition as the object ingredients get higher weights than the background ingredients.
> (2) The learned edge weights highlight the connections between the foreground object parts and the surroundings, which is consistent with human intuition.
> (3) Driven by sufficient instant graphs, the class graphs show the capacity to capture and model the general character of instances and eventually form the abstract imaginations of the categories.
>
> **Q3:** How would the authors explain why an instance IR-Graph (representing a particular input image) is similar to a class IR-Graph?
>
> **A3:** Thanks for your comments. We agree with the reviewer that a simple GCN is not interpretable, and we would like to clarify the interpretability of the proposed IR-Graphs.
> At the theoretical level, as shown in Theorem 1 and Corollary 1, the original complicated Equation 8 can be simplified by Equation 9, which eliminates cross terms, resulting in interpretable prediction results of how an instance graph is matched to a class graph.
> On the other hand, as shown in Figure 1, Figure 5, and Figure 12 to 14, the visualizations support the interpretability. Moreover, as illustrated in Figure 10, the misclassified examples clearly point out the critical points leading to the misclassification.
> Moreover, as in the extendability experiments in Appendix F.4, the GCN graph matcher is merely responsible for evaluating the similarity of any two graphs regardless of the tasks, meaning that no category knowledge is stored in the GCN.
>
> We once again thank the reviewer for the insightful comments. We hope our responses and revision have addressed your concerns. Please let us know if you have other questions. We are happy to further answer the questions.
>
> **References**
>
> [1] Evaluating bag-of-visual-words representations in scene classification, MIR 2007.

---

> > ### Comment · Reviewer_bkvM · 2022-12-06
> > **Thank you for your response**
> >
> > Thank you for your response. I am maintaining my overall positive assessment of the paper.

---

### Official Review · Reviewer_BvuD · 2022-10-28

**Confidence:** 5
**Correctness:** 4
**Technical Novelty And Significance:** 3
**Empirical Novelty And Significance:** 3
**Recommendation:** 6

**Clarity, Quality, Novelty And Reproducibility:**

Overall, the paper is well written, and addresses an interesting and important problem. The proposed method makes sense intuitively based on the inspiration of the cognitive concept schema. Source code is provided but not checked by the reviewer.

**Strength And Weaknesses:**

Strength:

1) The proposed method leverages the ensemble-level information (the codebook and the category IR-Graphs), as a post-hoc processing step to address explainable image classification.
2) The proposed method is able to explain the image classification results in terms of graph visualizations.
3) The proposed method obtains promising results on three datasets (CIFAR10/100 and Caltech 101).

Weaknesses:

1) This paper aims to address explainable image classification. There are a few aspects that could be addressed to consolidate the proposed method.  It will be better to show the inference results of misclassified examples to check if the inferred IR-Graphs can indeed provide insights of why they are misclassified.  It will also be interesting to compare with the recent methods explaining ViT models (e.g., Chefer  et al, Transformer interpretability beyond attention visualization, CVPR21). It will also be important to see if the proposed inference method is potential more robust to adversarial attacks as a sanity-check of the targeted explainability.

2) Since the foundation of the proposed method is built on the codebook computed via k-mean clustering running on the collection of visual tokens extracted by a feature backbone (DEIT) from a probe dataset, it will be better to show the learned mean image patches to see if they make sense visually.  Individual image patches are shown in the IR-Atlas (e.g. Fig.5). How are they selected to represent each codebook index?
It will also lead to the potential scalability concerns.  The proposed method is tested on relatively small datasets, rather than the ImageNet-1K. It will be interesting to see ImageNet results if possible.  How large will the codebook need to be, as well as the training and inference cost  of the SchemaNet?

3) The proposed graph similarity computation via GCN is not clearly explained. Vertex embeddings are used and trained by GCNs. It seems that the backbone only provide initialization of the IR-Graph structures. The resulting explainablity may thus have a gap with the backbone. Will it be possible to use the visual tokens as features for the vertex in the IR-Graphs?

**Summary Of The Paper:**

This paper presents the SchemaNet, an inference paradigm based on the cognitive concept of schema for inducing explainability of  trained image classification Vision Transformers. It first builds a codebook of by k-mean clustering running on the collection of visual tokens extracted from a probe dataset. With the codebook, it builds the ingredient relation graph (IR-Graph) at the instance level using a Feat2Graph module, and at the category level. It then exploits graph convolution network for computing the graph similarity of instance and category IR-Graphs for explainable classification inference. In experiments, the proposed method is tested using DEIT on CIFAR10/100 and Caltech-101 with promising results obtained.

**Summary Of The Review:**

Overall, the proposed method is an interesting and promising approach. The reviewer would like to see the rebuttal on the aforementioned weaknesses.

---

> ### Author Response · Authors · 2022-11-19
> **To Reviewer BvuD (2/2)**
>
> **Q6:** The proposed method is tested on relatively small datasets rather than the ImageNet-1K. It will be interesting to see ImageNet results if possible.
>
> **A6:** We thank the reviewer for encouraging us to perform more experiments on ImageNet-1k.
> Based on our experience, the number of ingredients should be set about ten times the category number, meaning 10^3 x 10^4 x 10^4 memory overhead for the learnable IR-Atlas, which is a major challenge.
> Despite this, we have tried our best to reduce the parameter size by supporting sparsity graphs so that we are able to conduct experiments on ImageNet-1k with 8000 ingredients, depending on the initialization.
> The results and detailed implementation of ImageNet-1k and mini-ImageNet are in Appendix F.1, showing consistent results with the main results in Table 2 in the main text: SchemaNet also achieves superior performance than the baseline methods, validating the effectiveness of our method on the large-scale dataset.
>
>
> **Q7:** How large will the codebook need to be, as well as the training and inference cost of the SchemaNet?
>
> **A7:** Thanks for raising the question of the selection of the codebook size, i.e., visual vocabulary size. We compare different sizes of the visual vocabulary on CIFAR-100 in Figure 8(a) in Appendix G. It can be observed that even with a relatively small size, e.g., M = 256, the performance of SchemaNet is still competitive (about 7% absolute degradation). However, for the case of M = 2048, the performance suffers from the low generalizability of the vocabulary. Hence we choose the medium-sized codebook (M = 1024) for CIFAR-100 to achieve better performance and generalizability. For the ImageNet-1k dataset, we use 8000 as the codebook size to balance the performance and the memory overhead. However, as conducting experiments on ImageNet-1k is expensive, we are not able to test performance with different sizes.
> As for the inference cost, we have listed the inference time of each component in Table 7. The running time of the bottleneck, i.e., the Feat2Graph module, is about 1.75 ms per image for about 100 classes.
>
>
> **Q8:** The proposed graph similarity computation via GCN is not clearly explained. The resulting explainability may thus have a gap with the backbone. Will it be possible to use the visual tokens as features for the vertex in the IR-Graphs?
>
> **A8:** Thanks for pointing out the concern. We have updated the manuscript in Section 2.3 and Section 3.4 to make clearer discussions on the interpretability of the visual words and the class IR-Atlas.
> In summary, the graph similarity between the instance and category graphs is dominated by the joint vertices that occur in both graphs, with two components: vertex importance and the similarity of the local structure of each critical vertex, which is supported by Theorem 1 and Corollary 1. Moreover, the visualizations in Section 3.4 and Appendix I support the interpretability.
>
> As for the visual tokens, they can't be used as features for the vertex due to the following two reasons. (1) Here, we want to highlight that the aim of our paper is to use the ingredient relationship for interpretable prediction instead of relying on deep features.
> (2) In fact, using randomly initialized vertex embedding is a crucial premise in Theorem 1, helping eliminate cross terms in Equation 8, which results in interpretability. Although the vertex embeddings are learnable, such property is still preserved, as shown in Figure 6. While employing deep features as vertices is not guaranteed to meet this assumption.
>
>
> We appreciate your time and insightful suggestions. We hope our responses and revision have addressed your concerns. Please let us know in case we have missed any open points. We are happy to further answer the questions.
>
> **References**
>
> [1] Transformer Interpretability Beyond Attention Visualization, CVPR2021
>
> [2] Learning Deep Features for Discriminative Localization, CVPR2016
>
> [3] On robustness and transferability of convolutional neural networks, CVPR2021
>
> [4] The many faces of robustness: A critical analysis of out-of-distribution generalization, ICCV2021

---

> > ### Comment · Reviewer_BvuD · 2022-11-21
> > **Response to Authors**
> >
> > Thank you to the authors for the rebuttal. Most of my concerns have been addressed. I changed my recommendation score to 6.

---

> > > ### Author Response · Authors · 2022-11-29
> > > **Thanks!**
> > >
> > > Thank you and the other reviewers for the valuable suggestions!

---

> ### Author Response · Authors · 2022-11-19
> **To Reviewer BvuD (1/2)**
>
> We thank the reviewer for the valuable comments, and we address the concerns and questions as follows:
>
> **Q1:** It will be better to show the inference results of misclassified examples to check if the inferred IR-Graphs can indeed provide insights into why they are misclassified.
>
> **A1:** Thank you for the valuable suggestion! We have added the visualization and discussions of the misclassified examples in Appendix I.3.
> Briefly, most misclassifications are caused by the noisy discretized ingredients, which may further form a remarkable local structure of the misclassified categories.
> Despite that, our proposed SchemaNet can highlight the key points leading to misclassification. The performance would be even better if the backbone DNN could be supervised to extract more robust local features, which we will leave to future works.
>
> **Q2:** It will also be interesting to compare with the recent methods explaining ViT models [1].
>
> **A2:** We thank the reviewer for pointing out the reference[1]. We have added more discussions with novel feature attribution methods for XAI in the related works in Appendix A.4. Moreover, we also compare the proposed SchemaNet with ViT-LRP [1] and CAM [2] in Appendix F.2. As shown in the following table, our method can achieve better performance using less running time compared to CAM and ViT-LRP. We explain these results as (1) the attribution methods compute a saliency map for each class (particularly, ViT-LRP conducts C-times backpropagation, consuming enormous time), which is significantly slower than our implementation; (2) further, as the backbone predicts soft probability distributions other than one-hot targets, the computed saliency maps for similar categories will be highlighted to some extent, impairing the graph matcher.
>
> ```
> |  Dataset  | Raw Attention |      CAM      |    ViT-LRP    |
> |           | Acc   | Time  | Acc   | Time  | Acc   | Time  |
> -------------------------------------------------------------
> | CIFAR-10  | 95.96 | 0.8h  | 93.58 | 3.6h  | 91.28 | 53.8h |
> | CIFAR-100 | 78.45 | 3.7h  | 77.55 | 15.3h |   -   |   -   |
> ```
>
>
> **Q3:** It will also be important to see if the proposed inference method is potentially more robust to adversarial attacks as a sanity-check of the targeted explainability.
>
> **A3:** Thank you very much for your constructive suggestions. We updated the manuscript and added the experiments of adversarial attacks in Appendix F.3. We evaluate the baseline methods and the pre-trained SchemaNet (from ImageNet-1k described in Appendix F.1) on two popular adversarial benchmarks: ImageNet-A [3] and ImageNet-R [4], as listed in the following table. It can be observed that SchemaNet is more robust than the baseline interpretable approaches, verifying that our method can show more stable performance under adversarial attacks.
>
> | Dataset | Backbone-FC | BoVW-Deep | BagNet | SchemaNet-Init |
> | - | - | - | - | - |
> | ImageNet-A | 5.41 | 7.36 | 5.53 | 13.05 |
> | ImageNet-R | 31.21 | 24.17 | 30.93 | 33.46 |
>
>
> **Q4:** It will be better to show the learned mean image patches to see if they make sense visually.
>
> **A4:** Thanks for your suggestions. Due to the sample diversity, images belonging to the same category will have different scales, occlusion, and angles in the dataset. For this reason, the mean image patches don't contain much information. Alternatively, we visualize the top-40 image patches closest to the cluster for each visual ingredient on the Caltech-101 dataset in Figure 11 in Appendix I.1. It can be easily seen that the obtained ingredients capture the most representative visual patterns in the given dataset, indicating that the visual words indeed represent meaningful and semantic-rich information.
>
>
> **Q5:** How are the individual image patches shown in the IR-Atlas (e.g., Fig.5) selected to represent each codebook index?
>
> **A5:** We thank the reviewer for proposing this question. In fact, they are excerpts of appeared ingredients in Figure 11 for quicker reference. We clarify this in Section 3.4.

---

### Author Response · Authors · 2022-11-19
**Summary of Revisions**

Dear reviewers and AC,

We sincerely appreciate your valuable time and constructive comments.
We have uploaded a revised version of our paper, where major changes are highlighted in blue. Below is the summary of major changes:

1. We revise Section 2.3, which may cause confusion about the GCN's computation and the interpretability of the graph matcher.

2. We add the evaluation of the interpretability in Section 3.3.

3. We provide more discussions and interpretations about the visualization of the IR-Atlas in Section 3.4, highlighting the similarity between machine-learned knowledge and human knowledge.

4. We add discussions about feature attribution methods in Appendix A.4.

5. We provide experimental results on ImageNet-1k in Appendix F.1.

6. We provide comparison results using different attribution methods in Appendix F.2.

7. We add experiments of adversarial attacks in Appendix F.3.

8. We add sensitivity analysis of the sparsification threshold in Appendix H.

9. We further provide visualizations of ingredients and analysis of misclassified examples in Appendix I.

We sincerely hope our responses and revisions address all reviewers’ concerns, and believe these updates may help us better deliver the benefits of the proposed SchemaNet to the ICLR community.

Thank you very much,

Authors.

---

### Decision · Program_Chairs · 2023-01-20

**Decision:**

Accept: poster

**Justification For Why Not Higher Score:**

The weak aspects pointed out by the authors and this final suggestion on adding more comparisons suggest the paper would need some more improvements to be accepted as spotlight.

**Justification For Why Not Lower Score:**

This paper presents an interesting new explainability approach and all the reviewers agree on the value of the contributions.

**Metareview: Summary, Strengths And Weaknesses:**

This paper presents the SchemaNet, a new post-hoc explainability technique for vision models based on the cognitive concept of schema. The main experiments are performed on CIFAR10/100 and Caltech-101.

The reviewers valued the interest of the SchemaNet and the obtained results. The reviewers made some comments and suggestions to improve the paper (e.g. comparison with other recent explainability methods; adding experiments on adversarial attacks; need for clarifications on the graph similarity computation via GCN; extending the explanation on how to interpret the learned class IR-Graphs, and what is the relation between an instance IR-Graph and a particular class IR-Graph; need for a deeper analysis of the obtained interpretability). The comments and suggestions were addressed by the authors in a satisfactory manner. Overall, the evaluations of this paper are positive and the reviewers agreed on accepting the paper.

In the final version of the paper authors need to address this last comment of one of the reviewers: "I would suggest to add the comparison of visualization to the other existing XAI methods (CAM, ViT-LRP), which can help highlighting the uniqueness of this method. Other approaches (CAM, ViT-LRP) mostly give attention based explanations, but this paper presents a new type of explanation. I find this part is novel but not sufficiently highlighted, and discussed".


**Note From Pc:**

if the above contains the word "oral" or "spotlight" please see: "oral" presentation means -> notable-top-5% and "spotlight" means -> notable-top-25%. As stated in our emails, we are disassociating presentation type from AC recommendations